# Gut microbial structural variation associates with immune checkpoint inhibitor response

Rong Liu [1,2,3,4] ✉, You Zou[5], Wei-Quan Wang[1,2,3,4], Jun-Hong Chen[1,2,3,4], Lei Zhang[1,2,3,4], Jia Feng[1,2,3,4], Ji-Ye Yin [1,2,3,4], Xiao-Yuan Mao[1,2,3,4], Qing Li [1,2,3,4], Zhi-Ying Luo[6,7], Wei Zhang [1,2,3,4] ✉ & Dao-Ming Wang [8,9] ✉

The gut microbiota may have an effect on the therapeutic resistance and toxicity of immune checkpoint inhibitors (ICIs). However, the associations between the highly variable genomes of gut bacteria and the effectiveness of ICIs remain unclear, despite the fact that merely a few gene mutations between similar bacterial strains may cause significant phenotypic variations. Here, using datasets from the gut microbiome of 996 patients from seven clinical trials, we systematically identify microbial genomic structural variants (SVs) using SGV-Finder. The associations between SVs and response, progression-free survival, overall survival, and immune-related adverse events are systematically explored by metagenome-wide association analysis and replicated in different cohorts. Associated SVs are located in multiple species, including *Akkermansia muciniphila*, *Dorea formicigenerans*, and *Bacteroides caccae*. We find genes that encode enzymes that participate in glucose metabolism be harbored in these associated regions. This work uncovers a nascent layer of gut microbiome heterogeneity that is correlated with hosts' prognosis following ICI treatment and represents an advance in our knowledge of the intricate relationships between microbiota and tumor immunotherapy.

Immune checkpoint inhibitor (ICI) immunotherapy has revolutionized the area of tumor therapy and resulted in remarkable advancements in the therapy of malignancies. Programmed cell death 1 (PD-1), programmed cell death 1 ligand 1 (PD-L1), cytotoxic T lymphocyte-associated antigen-4 (CTLA-4) and other targets are specifically targeted by ICIs in order to effectively release immunological brake reactions and suppress tumor immune escape. Antibodies such as ipilimumab, pembrolizumab, and nivolumab are used as initial therapies for a range of malignancies, including melanoma[1], and gastric cancer[2], demonstrating an exceptional increase in patient survival.

However, it is noteworthy that responses to ICI therapy exhibit heterogeneity, with response rates ranging from 13 to 69%[3]. Several factors have been identified to affect the effectiveness of ICIs, including the tumor surface's PD-L1 expression level[4], the tumor mutational burden[5,6], and the activity of interferon-γ pathway[7]. Given this, altering the gut microbiota offers a potentially useful method of augmenting the antitumor immune response and broadening the effectiveness of ICIs.

The functions of the microbiota and its metabolites in influencing immune reactions locally or systematically have garnered considerable

[1]Department of Clinical Pharmacology, Xiangya Hospital, Central South University, 87 Xiangya Road, Changsha 410008, P. R. China. [2]Institute of Clinical Pharmacology, Central South University, Hunan Key Laboratory of Pharmacogenetics, 110 Xiangya Road, Changsha 410078, P. R. China. [3]Engineering Research Center of Applied Technology of Pharmacogenomics, Ministry of Education, 110 Xiangya Road, Changsha 410078, P. R. China. [4]National Clinical Research Center for Geriatric Disorders, 87 Xiangya Road, Changsha 410008 Hunan, P.R. China. [5]Information and Network center, Central South University, Changsha 410083, P.R. China. [6]Department of Pharmacy, The Second Xiangya Hospital, Central South University, Changsha, PR China. [7]Institute of Clinical Pharmacy, Central South University, Changsha, PR China. [8]University of Groningen, University Medical Center Groningen, Department of Genetics, Groningen 9713AV, the Netherlands. [9]University of Groningen, University Medical Center Groningen, Department of Pediatrics, Groningen 9713AV, the Netherlands. ✉e-mail: liuronghyw@csu.edu.cn; csuzhangwei@csu.edu.cn; d.wang@umcg.nl

attention in the context of cancer-immune system interplay and therapeutic response to ICIs[8–10]. The gut microbiota has been shown through a growing body of preclinical and clinical evidence to have the ability to influence antitumor immunity and impact the effectiveness of ICIs in managing melanoma, renal cell carcinoma (RCC), and non-small cell lung cancer (NSCLC)[11–19]. In mouse models, gut microbiota composition was found to affect responses to anti-PD-L1 inhibitors, with differences in responses eliminated through fecal microbial transplantation (FMT) or cohousing. Dendritic cell maturation was enhanced and CD8 + T cell priming was increased by the oral administration of *Bifidobacterium*, restoring the PD-L1 blockade's anticancer effectiveness[14]. In mice with melanoma, gavage with *Bacteroides fragilis* improved anti-CTLA-4 therapy effectiveness[19]. Patients with more diverse bacterial populations in NSCLC and RCC were found to be more responsive to PD-1-based immunotherapy. Oral supplementation of mice with *Akkermansia muciniphila (A.muciniphila)* after FMT improved the anticancer effects of PD-1-based immunotherapy in ICI non-responders[18]. The predictive effectiveness of *A.muciniphila* was validated in a prospective clinical study of NSCLC subjects following PD-1 inhibitor treatment in 2022[20]. Patients with melanoma who responded to PD-1-based immunotherapy showed higher levels of relative *Faecalibacterium prausnitzii* abundance as compared to those who did not react to the immunotherapy[21]. Another indicator of responsiveness to anti-PD-1 blockade was a higher abundance of a collection of eight species driven by *Bifidobacterium longum*[17], and a high proportion of *Bacteroides caccae* was typical in patients who were sensitive to ICI immunotherapy[22]. *Bifidobacterium pseudocatenulatum*, *Roseburia spp.*, and *A. muciniphila*, were discovered to be a panel of species that were correlated with the therapeutic sensitivity of ICIs[23]. In clinical studies, FMT treatment resulted in beneficial modifications to immune cell infiltrates in the intestinal lamina propria and tumor microenvironment[13]. Metabolites identified as one of the primary mechanisms, which are small molecules that can disseminate from the gut to influence both local and systemic anticancer immune reactions, enhancing the efficiency of ICI.

Highly variable sections of bacterial genomes, termed microbial structural variants (SVs), can be discovered from metagenomic sequencing data[24]. SVs consist of deletion SVs (dSVs), which are deleted from certain species, and variable SVs (vSVs), which differ in the number of copies among species. Microbial SV regions may include genes that are involved in interactions between the host and the microbe; as a result, they may be able to provide details on the resolution of bacterial functioning at the sub-genome level. A number of correlations have been observed between microbial SVs and the blood biochemical parameters of the host, including HbA1c, glucose, and total cholesterol[24]. Additionally, recent research has also discovered connections between gut microbial SVs and metabolites levels in host's blood, linking genetically encoded functions of bacteria with metabolites, and supplying possible molecular insights for the functional output of the microbiome[25]. Specifically, associations have been reported between bile acids and microbial SVs, and bacterial genes linked to host bile acid metabolism or indirectly involved in the alteration of primary bile acids have been identified[26,27]. Furthermore, it was discovered that SVs of *Bifidobacterium* and *Enterococcus* coordinate the metabolomic perturbations in essential congenital heart disease[28]. Despite these findings, whether SVs in the gut microbiome contribute to ICI responses through metabolomics still lack of investigations.

This study aimed to assess the relationships between the SV of the gut microbiome and the response to ICI therapies in a systematic manner. With the usage of a total of 996 patients from seven independent cohorts (Fig. 1), systematic microbial SV association analysis between response, progression-free survival (PFS) at 12 months, overall survival (OS), and immune-related adverse events (irAEs) of patients following ICIs treatment and SVs was conducted. We find associated SVs be located in multiple species, such as *Akkermansia muciniphila*, *Dorea formicigenerans*, *Bacteroides caccae* and *Alistipes shahii*. Moreover, some genes that encode enzymes that participate in glucose metabolism be harbored in the genome region of these associated SVs. The analysis led to the identification of putative

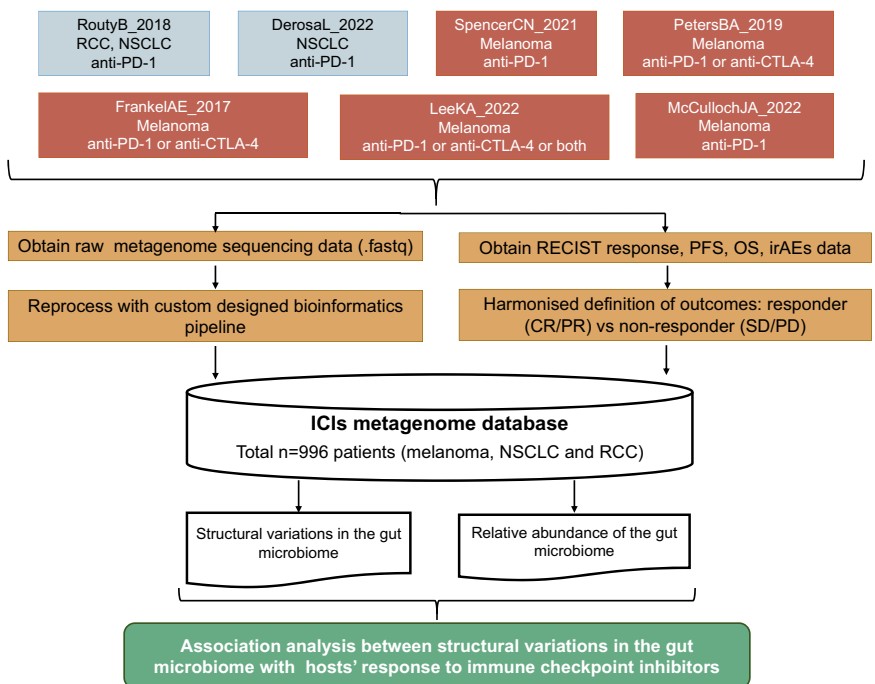

**Aggregation of publically available immune checkpoint inhibitor studies with metagenome data:**

**Fig. 1 | The layout of this study.** Design of this study. NSCLC non-small cell lung cancer, RCC renal cell carcinoma, OS overall survival, PFS progression-free survival, irAEs immune-related adverse events. RECIST response evaluation criteria in solid tumors.

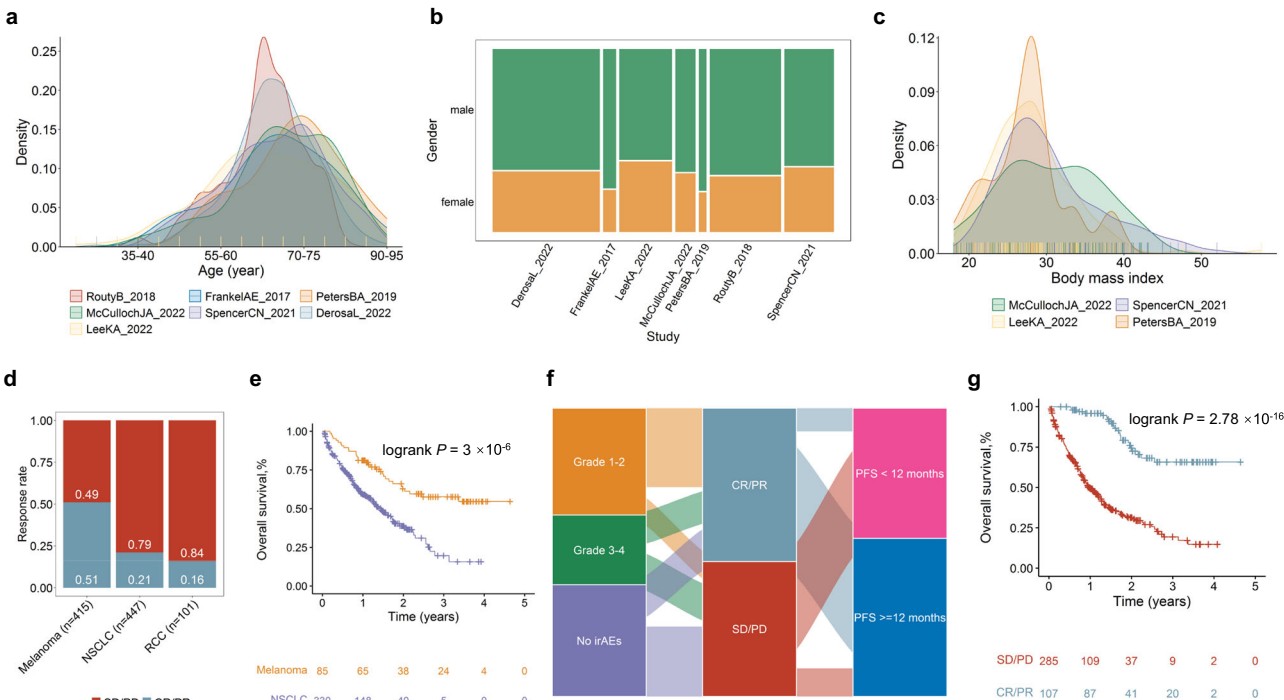

**Fig. 2 | Clinical characteristics of ICI related datasets. a** Age distribution in these seven studies. **b** Gender proportions of these seven studies. **c** Body mass index distribution in four studies with this information available. **d** Spine plots for response "CR/PR" versus no response "SD/PD". **e** Survival curves for OS by cancer types. **f** Sankey plot for irAEs, response, and whether progression-free survival longer than 12 months. **g** Survival curves for OS of response "CR/PR" and no response "SD/PD" groups. *P* values from log-rank tests are shown in survival plots. irAEs immune-related adverse events, PFS progression-free survival, OS overall survival.

microbial SV that affect ICIs medication efficacy, thus indicating the potential for manipulating the gut microbiome to enhance the effectiveness of ICIs treatment.

## Results

### Response to ICIs of cohorts

In this study, we collected classical clinicopathologic factors, including age, gender, response, PFS12, and OS, from seven cohorts (Fig. 2, Supplementary Data 1B). We calculated and compared the ratios of response to ICI therapies across cancer subtypes (Fig. 2d) and found that melanoma had the highest response ratio to ICI therapy (51%), whereas RCC had the lowest response ratio (16%). The average PFS12 rate is 48.8% in melanoma. We observed that melanoma patients demonstrate a higher OS rate than those with NSCLC (Fig. 2e, log rank *P* = 0.001) treated with ICIs. We also visualized the relationships between irAEs, responses to ICIs, and PFS12 using Sankey diagrams (Fig. 2f). Patients who responded to ICIs had considerably greater OS rates compared with non-responders (Fig. 2g, HR = 0.09, 95% CI = 0.05 - 0.14, log rank *P* < 2 × 10⁻¹⁶).

### Bacterial SVs identification

After filtering the SVs with abundance unavailable, 6715 SVs in 54 microbial species genomes, consisting of 1948 vSVs and 4767 dSVs, with 23–317 SVs per species, were detected in the UK cohort (Fig. 3a, b; Supplementary Data 2A). These 54 species, which ranged in microbial composition from 2.45 to 83.60%, made up an average of 46.54% of the total microbial composition (Fig. S2a). The 54 species had an average of about 51 patients with sufficient coverage to be called microbial SVs (Fig. S2b; Supplementary Data 2A), with *B.wexlerae, Collinsella sp, B.longum,* and *B.wexlerae* being the most commonly detected bacterial species. Meanwhile, 6,499 SVs in 49 microbial species genomes, consisting of 2,118 vSVs and 4,381 dSVs, with 38 to 354 SVs per species, were detected in the France cohort (Fig. 3e, f; Supplementary Data 2B).

These 49 species, which ranged in microbial composition from 5.56 to 85.17%, made up an average of 46.75% of the total microbial composition (Fig. S3a). The 49 species had an average of 98 patients with sufficient coverage to be called microbial SVs (Fig. S3b; Supplementary Data 2A), with *B. uniforms, P.distasonis,* and *F.prausnitzi* being the most commonly detected bacterial species.

We calculated the distance of bacterial SV profiles as described in the method between all samples within the USA or UK cohorts for melanoma (Fig. 3c). Microbial abundance (the top five PCs) can explain about 7.55% of the variance in the metagenome-wide SV profile (*P*PERMANOVA = 0.001; Fig. S2c). After correcting for microbial abundance, the cohort contributed to SV differences (explaining 5.17% of the SV variance, *P*PERMANOVA = 0.001; Fig. S2c). Further, after correcting for microbial abundance, the SV principal coordinates (PCo) 1 and PCo2 demonstrated differences between these five cohorts (ANOVA test, *P* = 1.35 × 10⁻⁹ for PCo1 and *P* = 1.34 × 10⁻⁷ for PCo2), demonstrating a divergence of microbial SVs between these five cohorts that was independent of differences in their microbial abundances. It's interesting to note that age bins and sex, combined could only account for 0.67% of the SV profile variance in the USA and UK cohorts (Fig. S1c).

As for France cohorts for NSCLC or RCC, microbial abundance (the top five PCs) can explain about 6.11% of the variance in the metagenome-wide SV profile (*P*PERMANOVA = 0.001; Fig. S3C). The cohort explains 0.61% of the SV profile differences. After correcting for species abundance, there are differences between two cohorts from France of the PCo1 (ANOVA test, *P* = 0.03), and no differences was found of the PCo2 (ANOVA test, *P* = 0.42).

### Microbial SV associations to ICIs response are independent from taxonomic abundance

The correlations between the clinical outcomes of patients following ICI treatment and the relative abundance of bacterial species were

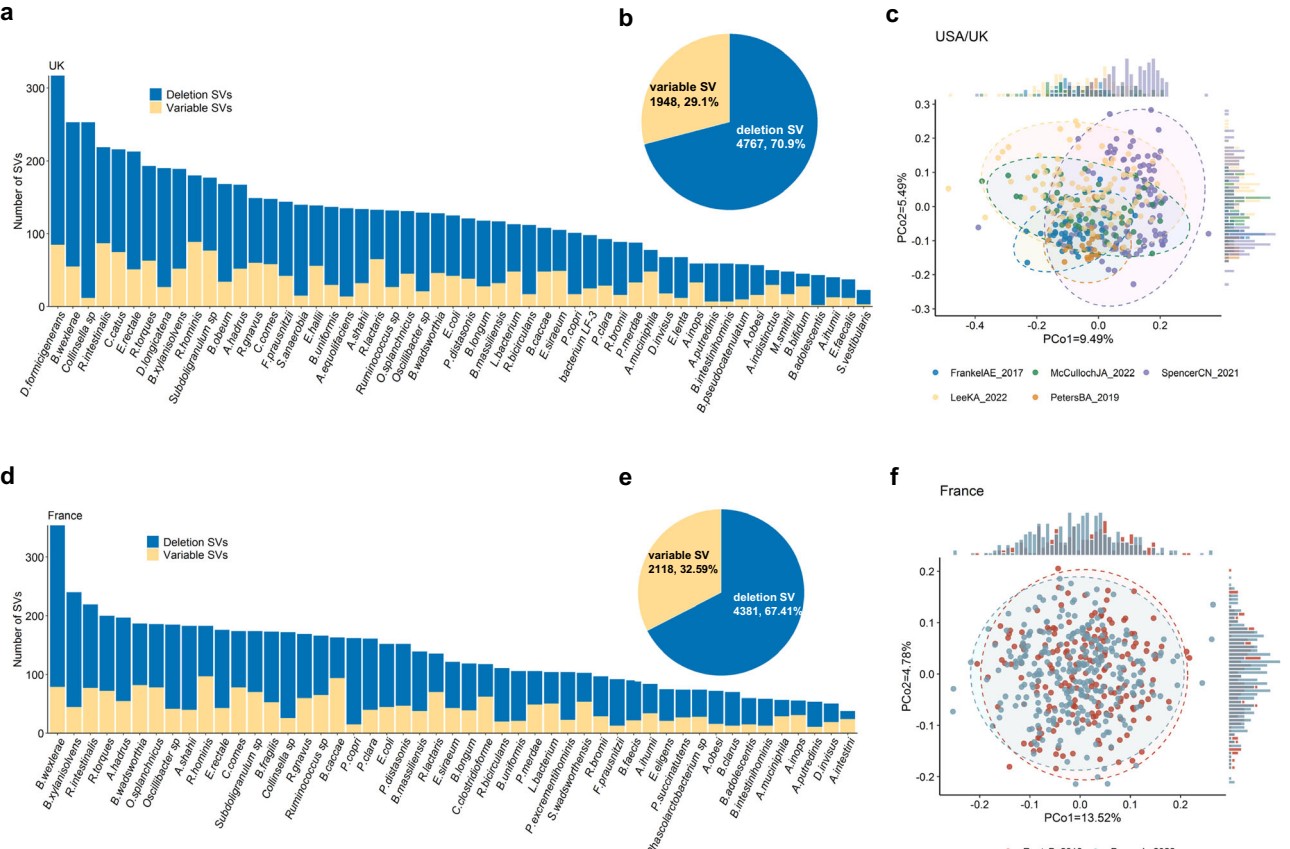

**Fig. 3 | Overview of structural variation profiles in the seven cohorts. a** The number of structural variations (SVs) present in each species within the UK cohorts from studies on melanoma. **b** The proportion of dSV and vSVs and total SV number of the UK cohort from studies on melanoma. **c** Principal component 1 and principal component 2 of SV makeup within five cohorts from USA or UK. **d** The number of SVs present in each species within the France cohorts from studies on NSCLC or RCC. **e** The proportion of dSV and vSVs and total SV number of the France cohorts from studies on NSCLC or RCC. **f** Principal component 1 and principal component 2 of SV makeup within two cohorts from France.

evaluated (Fig. 4). 48 significant associations, including 31 bacterial species based on our investigation of species abundance, were identified ($P_{meta} < 0.05$; Fig 4a, b; Supplementary Data 3). Among the associations, the most significant association was observed between the abundance of *B. wexlerae* and response to ICIs in NSCLC ($OR_{meta} = 1.11$, $P_{meta} = 4.86 \times 10^{-5}$; Supplementary Data 3D). Our findings confirmed previous research, including the positive association of the abundance of *D. formicigenerans* with the response to ICIs ($OR_{meta} = 1.08$, $P_{meta} = 3.06 \times 10^{-2}$; Supplementary Data 3A), PFS12 ($OR_{meta} = 1.14$, $P_{meta} = 5.17 \times 10^{-4}$; Supplementary Data 3B), and irAEs ($OR = 1.31$, $P = 1.77 \times 10^{-3}$, Supplementary Data 3C) of melanoma, which were consistent with the results of Frankel et al. [22]. Furthermore, positive associations between *R. bromii* and response to ICIs of NSCLC ($OR_{meta} = 1.06$, $P_{meta} = 1.87 \times 10^{-2}$, Supplementary Data 3D) were observed, which is also consistent with the previous findings[20].

In addition to the relative abundance of species, the SV of species is associated with therapeutic resistance and toxicity in patients treated with ICIs as well. A SV-based populational structure of the SV makeup for each species was constructed, and 47 significant associations between response/PFS12/irAEs and the SV makeup of 33 bacterial species were identified (meta $P_{PERMANOVA} < 0.05$; Fig. 4a, b; Supplementary Data 3), after accounting for confounding factors including age, and gender. It's interesting to note that just 9 of the 47 associations with species-specific SVs were found at the relative abundance level of species as well (Fig. S4, 7 of the 27 associations for the UK/USA cohort and 2 of the 20 associations for the France

cohort), underscoring the potential of microbial SV to provide additional information about bacterial functionality beyond species abundance. Our findings emphasize the significance of considering the SV of bacterial species as an important factor in contributing patient outcomes with ICIs.

Not only correlations at the species relative abundance level, but also the SV differences and relative abundance of *D. formicigenerans* were significantly correlated with response (Fig. 4c) and PFS12 of melanoma; meanwhile, the SV differences and abundance of *R. gnavus* (Fig. 4d), *A. shahii*, and *Ruminococcus sp* were associated with PFS12 of melanoma (Fig. 4a). Both the SV differences and abundance of *R. gnavus* and *B. wexlerae* were significantly associated with irAEs of melanoma (Fig. 4a). As for NSCLC, the abundance and SV differences of *D. invisus*, and *R. lactaris* (Fig. 4b) were significantly associated with response to ICIs. In addition, the SV difference of the other 12 species such as *B. adolescentis* (Fig. 4e) were associated with the responses to ICIs of NSCLC (Fig. 4f). The SV difference in *A. muciniphila* was associated with the responses to ICIs of melanoma and RCC (Fig. 4f). Furthermore, *R. bromii* was associated with the response of melanoma (Supplementary Data 3). We also found the SV of species be correlated with the prognosis after ICIs therapy, but not at the relative abundance scale. For example, SV profiles in *P. distasonis* associated with irAEs in melanoma ($P = 4.30 \times 10^{-2}$, Fig. 4a, Supplementary Data 3C), and response to ICIs in NSCLC ($P_{meta} = 1.69 \times 10^{-2}$, Fig. 4b, Supplementary Data 3D), but their abundance was not correlated with prognosis after ICI therapy. Our results suggest that species-specific SV makeup is

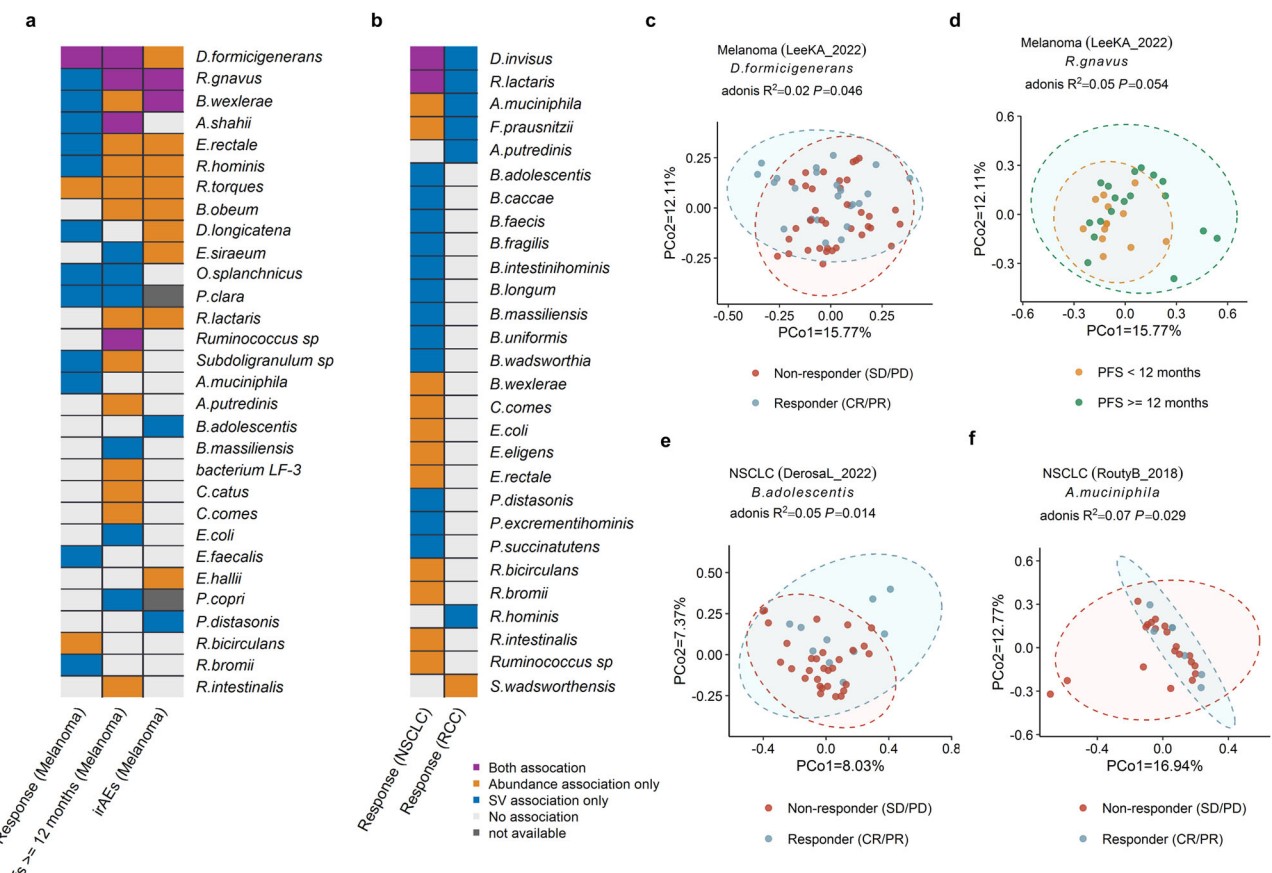

**Fig. 4 | Associations of gut microbiome with hosts' response to ICIs at the level of species.** Heatmap of associations between species and hosts' response to ICIs of melanoma (**a**), NSCLC and RCC (**b**). Yellow denotes purely relative abundance associations, blue indicates purely SV-based associations, and purple denotes associations based on both SV and relative abundance. Gray denotes relative abundance-based relationships, in which SV associations for the relevant species

were not examined due to the small sample size ($n < 10$). White denotes a lack of association. **c** SV association of *D.formicigenerans* with response of melanoma patients treated with ICIs. **d** SV association of *R.gnavus* with PFS12 of melanoma patients. **e** SV associations of *B.adolescentis* with response to ICIs of NSCLC. **f** SV association of *A.muciniphila* with response of RCC patients treated with ICIs. Two-sided statistical tests were utilized. Please also refer to Supplementary Data 3.

associated with ICIs drug responses independently of their taxonomic abundances.

## Metagenome-wide SV-based associations point to melanoma prognosis after ICIs treatment

In order to find SVs that contain genes potentially relevant to the response to ICIs, we conducted a microbial SV-based metagenomic analysis. To account for the potential heterogeneity within the cohort, we tested their associations with SVs separately in each of the studies used in this study, with age, and gender as covariates in the models, and then conducted meta-analysis to combine results when at least two cohorts exist for the same clinical outcomes. Since associations seen trending in the same direction or achieving significance in several individual studies are more compelling, given the evidence for reproducibility. Anywhere, to provide resources for reference, associations of clinical outcomes with more than one cohort available ($P_{meta} < 0.05$), and clinical outcomes with just one cohort available ($P_{normal} < 0.05$) were listed in Supplementary Data 4.

We have identified a total of 44 candidate SVs that are associated with clinical outcomes of melanoma within five cohorts across 23 species (Fig. 5; Supplementary Data 4A–4H). Among the species, *A.muciniphila* (Fig. 5b) demonstrated the highest number of associations, followed by *A.shahii*, *D.formicigenerans*, *Collinsella sp*, *A.putredinis*, *C.catus* and *P.distasonis* (Fig. 5c). We have depicted examples of significantly associated SVs in Fig. 5d–j. We found an YD repeat protein

encoded gene located in a dSV of *A. muciniphila* (863–865 kbp, Fig. 5d) was associated with both response to ICIs (OR$_{meta}$= 0.24; 95% CI = 0.0–0.61; $P_{meta} = 2.72 \times 10^{-3}$, Fig. 5e) and PFS12 (OR$_{meta}$ = 0.22, 95% CI = 0.08 – 0.58, $P_{meta} = 2.30 \times 10^{-3}$; Fig. 5f) of melanoma. Additionally, association was observed between response and the variable genomic segment (2,546-2,547 kbp, Fig.5g; OR$_{meta}$ = 1.65, 95% CI = 1.15 – 2.37, $P_{meta} = 7.01 \times 10^{-3}$, Fig.5h) of *A.muciniphila*, with gene *Amuc_2094* encodes glycosyl transferase family 2 and *Amuc_2095* encodes conserved hypothetical protein found to be within this region (Fig. 5g). Meanwhile, association was observed between response and the variable genomic segment (2547–2548; 2549–2550 kbp, Fig. 5g; OR$_{meta}$ = 1.72, 95% CI = 1.15–2.57, $P_{meta} = 8.75 \times 10^{-3}$, Fig. 5h) of *A.muciniphila*, with *Amuc_2096* encodes conserved polysaccharide biosynthesis and gene *Amuc_2097* encodes nitroreductase found to be within this region. We also found that a vSV with in *R.intestinalis* was associated with PFS12 (2891–2892 kbp: OR$_{meta}$ = 0.45, 95% CI = 0.26–0.80, $P_{meta} = 6.93 \times 10^{-3}$, Fig. 5j), with gene *RO1_28760* which encodes Relaxase/Mobilisation nuclease domain, and *RO1_28770* which encodes Bacterial mobilisation protein (MobC) found to be within this SV region. Associations between irAEs and 13 vSVs in *B.wexlerae* were identified (Supplementary Data 4F, fdr $P \le 0.1$). Nevertheless, due to the relatively small sample size ($n = 62$), these findings still need further validation. However, none of the SVs were found to be associated with OS of melanoma (fdr $P \le 0.1$, $n = 62$, Supplementary Data 3G, H).

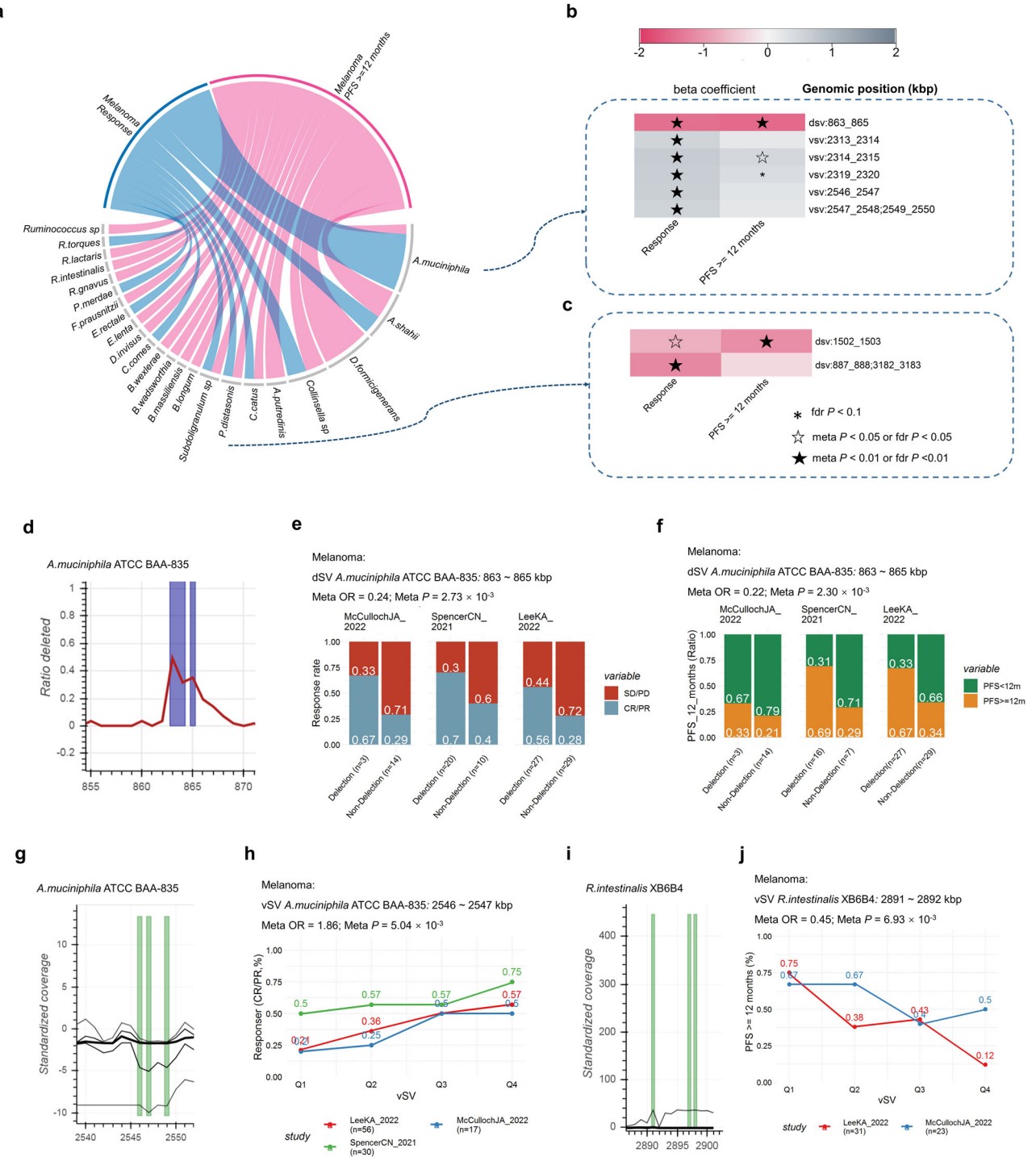

**Fig. 5 | Associations between structural variants and prognosis after ICIs treatment of melanoma patients. a** Candidate associations between prognosis after ICIs treatment and SVs of melanoma patients. Heatmap of correlations between prognosis after ICIs treatment and SVs of *A.muciniphila* (**b**) and *P.distasonis* (**c**). **d** Deletion rate across the cohort (y axis) along a genomic region of *A.muciniphila* (x axis). Spine plots depict the association between response (**e**), PFS12 (**f**), and dSVs within each cohort. **g** Standardized variability (y axis, plotted lines, percentiles 1, 25, 50, 75 and 99) along a genomic region of *A.muciniphila* (x axis). **h** Line plots depict the association between PFS12, and vSV within each cohort

(**h**). **i** Standardized variability (y axis, plotted lines, percentiles 1, 25, 50, 75 and 99) along a genomic region of *R.intestinalis* (x axis). **j** Line plots depict the association between response and vSV within each cohort, Q1 to Q4 are defined based on the quantiles of vSV (25%, 50% and 75%). Logistic regression models were performed to calculate beta value (**b**), ORs and 95% CIs (**e**, **f**, **h**, **j**) for response and PFS12. Meta-analysis with a random-effect model was performed to integrate the results of different cohorts. Two-sided statistical tests were utilized. No adjustments were made for multiple comparisons. Please also refer to Supplementary Data 4.

## Metagenome-wide SV-based associations point to NSCLC or RCC prognosis after ICIs treatment

Microbial SV-based metagenomic analyses were conducted for cohorts mainly from France, with cancer types of NSCLC or RCC. We

test associations between SVs separately in each of the 2 cohorts and then combine results with meta-analysis.

A total of 31 candidate SVs that are associated with clinical outcome across 15 species (Fig. 6a; Supplementary Data 4I–4N). *B.caccae*

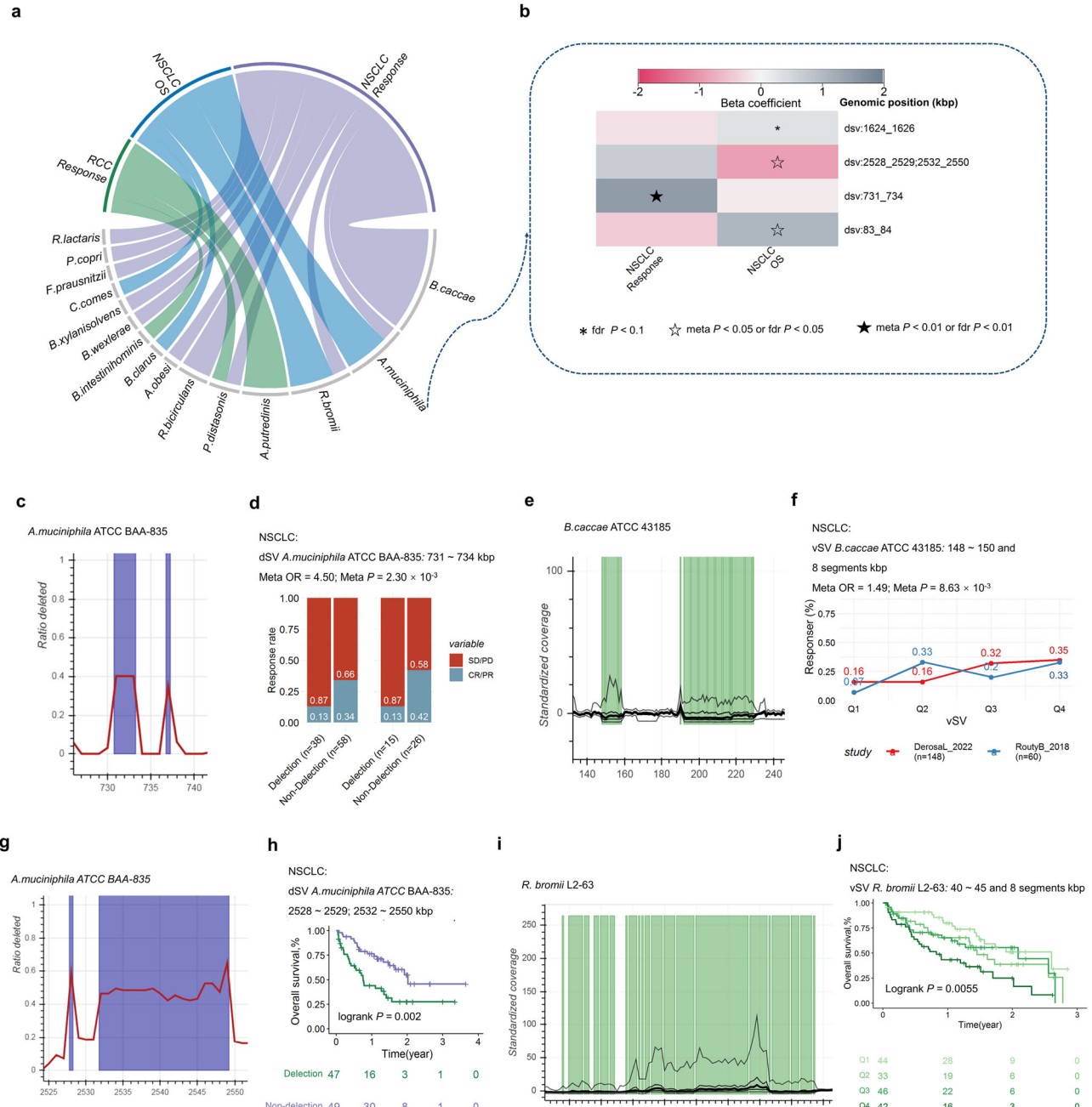

**Fig. 6 | Associations between structural variants and prognosis after ICIs treatment of NSCLC or RCC. a** Candidate associations between prognosis after ICIs treatment of NSCLC or RCC and SVs. **b** Heatmap of correlations between prognosis after ICIs treatment and SVs of *A.muciniphila*. **c** Deletion rate across the cohort (y axis) along a genomic region of *A.muciniphila* (x axis). **d** Spine plots depict the association between response, and dSVs. **e** Standardized variability (y axis, plotted lines, percentiles 1, 25, 50, 75 and 99) along a genomic region of *B.caccae*. **f** Line plots depict the association between response, and vSV within each cohort, Q1 to Q4 are defined based on the quantiles of vSV (25%, 50% and 75%). **g** Deletion rate across the cohort (y axis) along a genomic region of *A.muciniphila* (x axis).

**h** Survival curves of dSV and quartiles of vSV. **i** Standardized variability (y axis, plotted lines, percentiles 1, 25, 50, 75 and 99) along a genomic region of *R.bromii*. **j** Survival curves of vSV, Q1 to Q4 are defined based on the quantiles of vSV (25%, 50% and 75%). Illustrated *p* values are from log-rank tests. Logistic regression models were performed to calculate beta value (**b**), ORs and 95% CIs (**d**, **f**) for response. Cox regression models were performed to calculate beta value (**b**), ORs and 95% CIs (**h**, **j**) for response. Meta-analysis with a random-effect model was performed to integrate the results of different cohorts. Two-sided statistical tests were utilized. No adjustments were made for multiple comparisons. HR hazard ratio. Please also refer to Supplementary Data 4.

demonstrated the highest number of associations, followed by *A.muciniphila* (Fig. 6b) and R.*bromii*. Examples of significantly associated SVs were shown in Fig. 6c–j. Replicated association was observed between response and a dSV of *A.muciniphila* (731-734 kbp, Fig. 6c: $OR_{meta} = 4.50$, 95% CI = 1.71–11.85, $P_{meta} = 2.30 \times 10^{-3}$, Fig. 6d), with genes *Amuc_0622, Amuc_0623* encode glycosyl hydrolase BNR

repeat-containing protein, *Amuc_0624* encodes egulatory protein GntR HTH and *Amuc_0625* encodes Exo-alpha-sialidase found to be within this SV region. Further, a dSV at the genomic segments (4655–4656 kbp) of *P.distasonis* encoding a conserved hypothetical protein and putative nucleotide-sugar dehydrogenase that was significantly associated with response ($OR_{meta} = 0.39$, 95% CI = 0.22–0.68,

$P_{meta}$ = 8.94 × 10⁻⁴). Candidate-associated vSVs were also identified; for example, a vSV located in *B.caccae* (148-150 and 8 segments kbp, Fig. 6e) which includes 20 genes that encode enzymes like glycosyl hydrolase family 3 C-terminal domain protein, glycosyl hydrolase family 49, glycosyl hydrolase family 20, was associated with a better response (OR$_{meta}$ = 1.49, 95% CI = 1.11 - 2.01, $P_{meta}$ = 8.63 × 10⁻³, Fig. 6f). An association was found between the OS and a dSV of *A.muciniphila* (2528−2529; 2532−2550 kpb, Fig. 6g: HR = 0.39, 95% CI = 0.22 - 0.71, $P_{fdr}$ = 4.84 × 10⁻², Fig. 6h), with 18 genes found to be within this SV region that encode enzymes like glycosyl transferase families 1 and 2, polysaccharide biosynthesis protein, and nitroreductase. Four candidate vSVs in *R.bromii* were associated with OS, for example, a vSV (40 - 45 and 8 segments kbp, Fig. 6i) contains genes encoding ATPases involved in chromosome partitioning was associated with poor OS (HR = 1.38, 95% CI = 1.13 - 1.68, $P_{fdr}$ = 3.72 × 10⁻², Fig. 6j), and a vSV in *B.clarus* was associated with better OS (Fig. 6k). Three dSVs of *A.putredinis* were found to be associated with response to ICIs of RCC (Supplementary Data 4M), for example, detection of 107-109 kbp.

Notably, there were differences between cancer types in the effect sizes (OR or HR values) and directions of the associated relationships (Fig. S5).

## Discussion

A comprehensive analysis of the gut microbial SV in 996 individuals across seven independent cohorts globally was conducted. Our investigation involved a systematic evaluation of the associations between gut microbial species SVs and the host's reaction to ICIs. We discovered that the species SV profile correlates with the clinical outcomes after ICI treatment, independent of the species' abundances. To assess the association between microbial SV and response to ICIs, we carried out metagenome-wide microbial SV association analyses and identified candidate associations. Notably, this is the first investigation into the microbial SV determinants of response to ICIs in humans. We provide some clues for further mechanistic studies to explore how gut microbiota modulate antitumor immunity and affect the efficacy of ICIs.

Our investigation illustrates the effectiveness of the SV-based metagenome-wide association as a potent technique for understanding microbial associations at a functional and mechanistic level. Our study highlights that metagenomic SVs offer valuable data that explains the functionality of the human gut microbiome. Specifically, our research demonstrates that the associations between clinical outcomes after ICIs treatment and microbial SVs can remain independent of species relative abundances. We did not include the PCs of population SV structure in models when considering the influence of lineage effect on the metagenome-wide association study at the single SV level[25], as Wang et al. found that a model accounting for lineage effect reduces the statistical power for detecting some associations that involve SVs contributing to both clinical outcomes and bacterial genetic lineage[27]. Despite the fact that techniques developed for bacterial GWAS analysis were provided that take lineage effects and populational structure into consideration[29,30], these were designed for binary genetic variation data. Thus, the development of algorithms that adequately adjust for lineage effects for the metagenome-wide association study at the single SV level is still required.

Furthermore, our analysis revealed that the identified SVs were inclined towards extensively abundant and prevalent species, thereby emphasizing the challenges of investigating rare or low-prevalence species. This limitation calls for larger sample sizes and deeper sequencing to achieve adequate statistical power. Nonetheless, our sub-genomic investigation successfully identified the genomic regions that may associated with host response to ICIs, emphasizing the importance of linking microbial SVs from the entire metagenome with the diversity of phenotypes of human beings in locating microbial genetic elements or genes that contribute to host-microbe interactions. The gene *Amuc_0625*, which encodes an outer α-sialidase, is located on the genome fragment (731 to 734 kbp) of *A.muciniphila*. Heinz Läubli et al. have previously reported that macrophages polarize towards an M2 phenotype and produce an immunosuppressive response by recognizing sialic acid on the surface of tumor cells. They have also demonstrated that designing targeted drugs with bacterial sialidases can significantly enhance the efficacy of immunotherapy[31,32]. Effector T cells have been found to trigger IgG sialylation, which inhibits macrophage STING pathways, thereby weakening the efficacy of PD-L1 immunotherapy. The dSV of *A. muciniphila* contains α-sialidase, and may decrease its activity, thereby decreasing immune efficacy. Our investigation also identified genes encoding glucose metabolism-related enzymes on bacterial SV fragments, such as genes encoding glycosyl transferase families 1 and 2 located on genome fragments (2528 to 2529 kbp; 2532 to 2550 kbp; 2546 to 2547 kbp) of *A. muciniphila*, genes that encode enzymes like glycosyl hydrolase family 3 C-terminal domain protein, glycosyl hydrolase family 49, glycosyl hydrolase family 20 located in *B.caccae*, gene *PARMER_00338* encoding putative alpha-1,2-mannosidase on *P.merdae*, gene *PREVCOP_05967* encoding nucleotide sugar dehydrogenase on *Prevotella copri*, gene *BDI_3822* encoding putative nucleotide-sugar dehydrogenase on *P.distasonis*, gene *RBR_10280* encoding predicted ATPase on *R. bromii*, they roles in ICIs efficacy needs to be further explored. The gene *HMPREF1032_00306*, which encodes phage/plasmid primase, P4 family domain-containing protein, is located on the genome fragment (dSV: 2121 to 2122 kbp) of *Subdoligranulum*, deletion of this region is associated with lower response and PFS12 rates. Our comprehensive approach to association analysis establishes a framework for microbial SV studies.

Experimental validation of the causal relationship between microbial SV and ICI efficacy remains a daunting task, as it requires the development of a robust approach to prioritize genes within the SV regions. Moreover, it entails the isolation and cultivation of microorganisms from human feces samples, followed by sequencing to validate the existence or absence of SVs. Subsequently, oral administration of confirmed bacterial isolates with candidate SVs in mouse models was used to verify their impacts on the host's drug response to ICIs. However, each step encounters significant technical difficulties, particularly in isolating and culturing gut microbes, which remain elusive. Despite these challenges, our team is actively working to develop experimental platforms that can validate ICI-related SVs, and these efforts may eventually be helpful for better comprehension of the microbiome's function in ICI therapy.

We admit that our current study has a number of limitations. The samples utilized in this retrospective study were gathered from different countries worldwide, and the diversity of ICIs trials included in the analysis, which used various checkpoint blocking agents and varied combinations, increased the generalizability but constrained the specificity of our results. The sample size for association tests of clinical outcomes, such as OS and irAEs, per cancer type is small, which results in a lack of statistical power. The correlations between prognosis after ICIs therapy and microbial SVs call for validation in additional populations with a larger sample size. This study is a cross-sectional design that tests regulatory associations between responses to ICIs and microbial SVs. However, as we explained in the discussion, more confirmation in a longitudinal study cohort and through experimentation is still necessary to determine whether the changes in microbial genetic elements are causally related to the host's response to ICI medication treatment.

In conclusion, our study represents a significant advancement in the field of improvements to the host's responsiveness to ICIs by microbiome-targeted therapies. Through the analysis of relatively large datasets from seven ICI trials, we have contributed to a deeper understanding of the gut microbial SVs that are crucial for effective ICI therapy. This understanding will enhance our capacity to forecast and direct immunotherapeutic responsiveness and provide a route for more effective medication.

## Methods

### ICIs metagenomic sequencing datasets

This bioinformatics analysis utilized publicly accessible metagenomics sequencing data from ICI trials. All these studies have been previously approved by their respective institutional review boards. Raw gut metagenomic data was gathered from a range of research and cancer types, comprising a total of 996 ICI-treated patients from seven cohorts (Fig. 1, Fig.S1, Table 1, and Supplementary Data 1). No statistical method was used to predetermine sample size. Fig. S1 illustrates the sample filter process, samples didn't receive ICI treatment or collected after more than 4 months of from the start of ICI treatment, or without matched clinical and metagenomics data available were filtered from following analysis. To identify datasets, the initial author and year of publication were used. The gut metagenomics sequencing data from the FrankelAE_2017[22], McCullochJA_2022[33], RoutyB_2018[18], SpencerCN_2021[34], PetersBA_2019[35], DerosaL_2022[20], and LeeKA_2022[23] datasets were obtained from the ENA data portal (https://www.ebi.ac.uk/ena/browser/home). Clinical information was collected by searching the supplementary tables of the original articles. To maximize comparability across cohorts, we reprocessed these sequence data using a standardized bioinformatics pipeline.

### Definition of clinical outcomes

The study collected clinical information such as age, gender, ICI targets, PFS, and OS. While recognizing that the current definition of response is conservative and that patients who have stable disease (SD) and have extended survival can be thought of as experiencing clinical benefit from ICI treatment, we employed the following definition to ensure consistency with recent literature and clear response interpretation[5,36], which was determined using Response Evaluation Criteria in Solid Tumors (RECIST) criteria for radiological response as represented in the original articles. Responders were determined as patients who demonstrated a complete response (CR) or partial response (PR), while non-responders had SD or progressing disease (PD). The definition of progression-free survival at 12 months (PFS12) was that there was no disease progress as evaluated by RECIST 12 months after ICI treatment initiation. Response, PFS12, OS, and irAEs were utilized as clinical outcomes to ensure strict consistency in outcome measurement across the six studies (Table 1).

The abbreviations for clinical outcomes are listed as follows: PFS: progression-free survival; OS: overall survival; SD: stable disease; CR: complete response; PR: partial response; PD: progressing disease; PFS12: progression-free survival at 12 months; irAEs: immune-related adverse events.

### Metagenomic sequencing data preprocessing

The raw metagenomic sequencing data underwent a data cleaning procedure to remove low-quality reads and host genome-contaminated reads with the usage of KneadData (version 0.6.1), Trimmomatic (version 0.39)[37] and Bowtie2 (version 2.3.5.1)[38]. Briefly, the data preprocessing process involved two primary steps. Firstly, adaptor sequences and poor-quality reads were discarded using Trimmomatic (parameter settings: LEADING:3 TRAILING:3 SLIDINGWINDOW:4:20 MINLEN:50). Secondly, by mapping sequence data to the human reference genome (version GRCh37), human genome-contaminated reads were removed. The described data-cleaning procedure ensured the removal of low-quality and host genome-contaminated reads, thus enabling high-quality downstream analyses.

### Structural variations detection

Zeevi et al. [24]. have described SGV-Finder, a tool for detecting two kinds of SVs, namely dSVs and vSVs, from metagenomic sequence data. The SV-calling procedure can be mainly divided into two major steps: The first step involves running the iterative coverage-based read

**Table 1 | Summary of ICI metagenome studies**

| Study | Study name utilized in this study | Accession number | N# | N* | Cancer type included | Treatment | Clinical outcomes | Country$ |
|---|---|---|---|---|---|---|---|---|
| Frankel et al., Neoplasia 2017 | FrankelAE_2017 | PRJNA397906 | 39 | 39 | Melanoma | CTLA4/PD-1 blockade | Response | USA |
| McCulloch et al.,Nature Medicine. 2022 | McCullochJA_2022 | PRJNA762360 | 62 | 62 | Melanoma | PD-1 blockade | Response/irAEs/OS/PFS | USA |
| Spencer et al.,Science 2021 | SpencerCN_2021 | PRJNA770295 | 167 | 155 | Melanoma | PD-1 blockade | Response/PFS | USA |
| Peters et al., Genome Medicine 2019 | PetersBA_2019 | PRJNA541981 | 48 | 23 | Melanoma | CTLA4/PD-1 blockade | OS/PFS | USA |
| Lee et al.Nature Medcine.2022 | LeeKA_2022 | PRJEB43119 | 164 | 162 | Melanoma | CTLA4/PD-1 blockade | Response/PFS12 | UK |
| Routy et al., Science 2018 | RoutyB_2018 | PRJEB22863 | 219 | 219 | NSCLC; RCC | PD-1 blockade | Response | France |
| Derosa et al.Nature Medicine.2022 | DerosaL_2022 | PRJNA751792 | 338 | 336 | NSCLC | PD-1 blockade | Response/OS | France/Canada |

OS overall survival, PFS progression-free survival, NSCLC non-small-cell lung cancer, RCC renal cell carcinoma, PFS12 whether PFS longer than 12 months, irAEs immune-related adverse events.
\# Sample size in the original paper.
* Samples collected pretreated with ICIs and with matched clinical and metagenome data available which utilized in this study, the filter process listed in Fig. S1.
$ The country in which the study was recruited.

assignment (ICRA) algorithm, resolves ambiguous read with multiple alignments to regions that are comparable across different bacteria for the most likely reference in complex metagenomics settings based on the data of mapping quality, bacterial abundance, and genomic coverage. The second step involves running SGV-Finder, which splits the concatenated scaffolds from each microbial genome into 1 kbp bins and then analyzes the coverage in each metagenomic bins across all subjects to find highly variable genomic regions and identify SVs. Species with SV calls were absent in more than 95% of the whole samples filtered. If a SV were detected as both vSV and dSV, the dSV was kept. The proGenomes database (http://progenomes1.embl.de/) serves as the foundation for the reference database used by SGV-Finder.

SVs were detected based on the high-quality metagenomic sequence reads. Overall, we detected SVs across 164 samples (LeeKA_2022) from the UK using ICRA and SGV-Finder with default parameters (except --min_samp_cutoff 17). To analyze replication of associations between cohorts for melanoma, we calculated for each SV region in the UK cohort, its dSV or vSV in the FrankelAE_2017, McCullochJA_2022, PetersBA_2019, and SpencerCN_2021 cohorts from the USA. We run SGV-Finder with the --by-orig parameter by using the orig_dsgv.df, orig_vsgv.df and average coverage files (.df) file for each species generated from the UK cohort. Moreover, SVs detected across 338 samples (DerosaL_2022) from France were identified using SGV-Finder with default parameters (except --min_samp_cutoff 34). To analyze replication of associations between cohorts, we calculated for each SV region in the DerosaL_2022, its dSV or vSV in the RoutyB_2018 cohort, which is also from France, and ran SGV-Finder with the --by-orig parameter by using the orig_dsgv.df, orig_vsgv.df and average coverage file (.df) for each species generated from the DerosaL_2022 cohort. The min_samp_cutoff parameter was determined to be about 10% of the total sample size. The described approach enabled the detection of SVs, which may have implications for understanding the microbial community's structure and function.

### Taxonomic abundance
The taxonomic relative abundance of all samples utilized in this study was generated from high-quality metagenomic reads using Kraken2 (version 2.1.2)[39] and Bracken (version 2.6.1)[40]. The reference genomes were also developed based on the Progenome Database[41].

### Statistical analysis
Specific information regarding the statistical tests used can be found in the Results and the corresponding figure legends. Unless otherwise noted, a $P \leq 0.05$ was regarded as statistically significant. Two-sided statistical tests were utilized unless otherwise specified. Using the Kaplan-Meier method, survival curves were estimated, and the log-rank test was utilized to compare them. Cox regression models were conducted to calculate hazard ratios (HRs) and 95% confidence intervals (CIs) for OS. Logistic regression models were performed to calculate ORs and 95% CIs for response, PFS12, and irAEs. Meta-analysis with a random-effect model was performed to integrate the results of different cohorts.

All statistical tests were performed with R (version 4.0.5).

### Distance matrix-based variance estimation and principal coordinates analysis
In this study, we measured the variability of microbial vSVs between samples using the Canberra distance metric, and calculate the variability of microbial dSVs between samples using the Jaccard distance metric[42], then measure the variability of microbial SVs between samples using the average value of above two distance matrix. We accomplished this by computing the distance of SVs utilizing the vegdist() function from the R package vegan (version 2.5.6).

A principal coordinates analysis (PCoA) based on Canberra distance indices calculated with SVs were conducted utilizing the cmscale() function in the vegan. After that, to determine the ratio of SV profile variance that can be explained by factors such as microbial composition, age bins, gender, and different cohort, a permutational multivariate analysis of variance (PERMANOVA) with 999 permutations were performed using the Adonis() function from vegan.

### Association analysis
We examined the differences in response, PFS12, and irAEs following ICI treatment across the SV or abundance of species. The associations between OS and SV, or abundance, for each species were also investigated. The assignment values of the clinical variables were listed in Supplementary Data 1C.

### Species-level associations of the gut microbiome with clinical outcomes
The association between binary clinical outcomes and SV of each species was evaluated using PERMANOVA with 999 permutations with the following formula:

$$Distance\ matrix\ of\ SV \sim clinical\ outcome + Age\_bins + Gender \quad (1)$$

The association between binary clinical outcomes and species relative abundance was evaluated using a logistic regression model with the following formula:

$$Clinical\ outcome \sim Species\ relative\ abundance + Age\_bins + Gender \quad (2)$$

For each clinical outcomes, association analysis were performed within each cohort, and meta-analysis with a random-effect model was performed to integrate the results of different cohorts.

### dSV or vSV site based associations of the gut microbiome with clinical outcomes
Associations between SVs and binary clinical outcomes were assessed using logistic models with the formula:

$$Clinical\ outcome \sim SV + Age\_bins + Gender \quad (3)$$

, demanding at least 10 subjects in each comparison and at least 3 responders.

Associations between SVs and OS were assessed using Cox regression models with the formula:

$$OS \sim SV + Age\_bins + Gender \quad (4)$$

ensuring a minimum of 20 subjects in each comparison.

In order to derive more easily interpretable HRs or ORs, quartiles (25%, 50%, and 75%) of the value of each vSV were computed for association analysis and modeled as continuous variables.

The Benjamini-Hochberg (false discovery rate: FDR) P value correction method was applied with the p.adjust() function in R. Specifically, for vSVs, dSVs, and species relative abundance, we carried out association analysis and P value correction independently. If there is only one dataset for clinical outcome, we considered the SV-prognosis candidate associations with an FDR p value ≤ 0.1. If there is more than one dataset for clinical outcome, the replicated candidate associations were confirmed with the following three criteria: (1) $P_{meta} \leq 0.01$; (2) $P_{heterogeneity} > 0.05$; (3) Normal $P \leq 0.2$ within at least two cohorts and trending in the same direction (Supplementary Data 1D).

We calculated the Spearman's correlation coefficient between effect sizes within different cancer types for the analysis presented in Figs. S5A–S5D.

**Reporting summary**

Further information on research design is available in the Nature Portfolio Reporting Summary linked to this article.

## Data availability

Raw metagenomics sequencing data of the seven datasets are publicly available from the European Nucleotide Archive (https://www.ebi.ac.uk/ena/browser/home) via accession numbers (PRJNA397906, PRJNA762360, PRJEB22863, PRJNA770295, PRJNA751792, PRJNA541981 and PRJEB43119).

## Code availability

R scripts demonstrating how to reproduce all findings shown in the main figures are available via https://github.com/liuronghyw/ICIs_gut_microbe_SVs . Rong Liu, You Zou, Wei-Quan Wang, Jun-Hong Chen, Lei Zhang, Jia Feng, Ji-Ye Yin,Xiao-Yuan Mao, Qing Li, Zhi-Ying Luo, Wei Zhang, Dao-Ming Wang. Gut microbial structural variation associates with immune checkpoint inhibitor response, "ICIs_gut_microbe_SVs", https://doi.org/10.5281/zenodo.10020240, 2023.

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

## Acknowledgements

This study was partly supported by the National Key Research and Development Program 2021YFA1301200 (W.Z.), the Hunan Provincial Science and Technology Innovation Plan Project 2022RC1022 (R. L.), the NWO-Veni grant VI.Veni.222.016, (D.M.W.), the National Scientific Foundation of China 82373961 (W.Z.) and 31801121 (R.L.), and the key project of the Hunan Education Department 21A0002 (R.L.). The authors acknowledge the authors from published studies to share their metagenomics sequencing data of immunotherapy trials and are grateful for resources from the High Performance Computing Center of Central South University and The Bioinformatics Center, Xiangya Hospital, Central South University.

## Author contributions

R.L. conceived and supervised the study. R.L. contributed to the study design, performed statistical analysis, interpretation and drafted the manuscript. D.M.W. provided the pipeline (https://doi.org/10.5281/zenodo.5599104), also gave suggestions and discussions for the data analysis method. Y. Z. contribute to the data collection and provide assistance for parallel computing. All authors contributed to critical revision of the final manuscript. R.L., W.Z. and D.M.W. approved the final version of the manuscript.

## Competing interests

The authors declare no competing interests.
