## [Peer Review File · Nature Communications]

REVIEWER COMMENTS

Reviewer #1 (Remarks to the Author):

Liu et al. present an analysis of gut microbial abundances and structural variants in the context of response to immune checkpoint inhibitor therapy. They have harmonized a large volume of data (>1K participants) and obtained interesting results. This is an interesting problem and the premise for this work is very compelling. SV analysis could be very interesting in this realm and I think the results, especially at the end of the manuscript + interpretation in the discussion, showcase this potential. I think the manuscript is timely, interesting, largely well written, and the figures are beautiful. Well done to the authors.

I unfortunately have significant issues with some of the basic data analysis approaches taken here. I apologize in advance for quite the lengthy review – I really wanted to lay out all the different issues with the intent of helping the authors make this an excellent paper.

Major comments:

1. The authors generally take the approach of analyzing all the studies together. I think this can cause major issues. What I am most concerned about is that SVs are called on all studies together. Since SV “detection” is threshold-ed on passing some level of variability (deletions, for example, have to be deleted in between 25-75% of the samples), this actually biases SGVFinder to find strain differences between the cohorts, rather than to find regions that are truly adaptive within a cohort.

a. How many of the SVs have a significant difference between the studies?

b. This is probably my most important comment on this paper: Since this issue involves feature construction and not just data analysis, I strongly believe that just including the study in the regression model is not going to be enough. I think the best analytic approach would be to identify SVs within each study and then check for generalization (i.e., take L392-393 and the supp tables, and make them the focus of the manuscript), but I understand that this loses power significantly. In lieu of that, I think the authors should do a cross-study validation – i.e., detect SVs and their associations within 6 of the studies, and then check whether the associations replicate within the same study when SVs in that study are called with SGVFinder’s “by_orig” parameter. Then repeat this procedure for all studies. Then do downstream analyses only on SVs that truly generalize across studies.

c. The fact that the authors find varying enrichment in different cohorts (S4, L373-375) strongly supports this concern.

2. Association analysis:

a. Why would the authors include read count as a confounder? It seems that by construction SVs should not be affected by read depth, and the analysis in Fig. S1c actually supports that.

b. Why would the authors include species relative abundance in the model? That also seems unrelated for largely the same reasons. Is there an association between SVs of a microbe and the relative abundances of the same microbe? (not this is not the same as the association with microbiome PCs shown in Fig. S1C, unless something is very unclear in the description of the regression model).

3. Distance calculations:

a. I find it very odd to combine together continuous features (vSVs) and binary features (dSVs). Perhaps the authors should calculate one distance for each and average them.

b. The authors should justify or give intuition for the use of the Canberra distance.

c. What does “distance of shared SVs” (L240) mean? Is there a different dimensionality per sample pair? That could cause substantial issues in data analysis. If not – how are missing data handled?

4. Clustering analysis within species:

a. I don't understand the premise of this analysis. Is this supposed to be capturing sub-species information? Wouldn't mOTUs2 or MetaPhlan4 be more suited for this purpose? The authors should benchmark their explanatory power if this is their intent. (Running Kraken on Progenomes1 wouldn't cut it here).

I should note that I think this analysis misses the point of SVs which is to narrow down to a specific genomic region.

b. The associations in Fig. 4B-G seem extremely weak. Is this even significant? Was this adjusted for multiple testing? P-values are not shown anywhere (or written in the text).

c. Is the model written out in L268 applied in the permanova somehow, or is this unadjusted? It's very problematic if unadjusted.

Other comments:

1. The authors mention that they preferred dSVs to vSVs – why? I think both should be kept, and in any case, vSVs should have more resolution.

2. The sentences in L243-247 are unclear. What do the authors mean by “PCs [...] were chosen”? What was done with them? Later in the discussion (L451-453) they seem to imply that you didn't do anything with them?

3. How was missing data handled in general in the analysis? Did the authors only compare dSVs / vSVs within samples in which the species was identified?

4. The authors should specify the parameters for running SGVFinder. “min_samp_cutoff” is of particular interest here.

5. L259: A cutoff of 0.55 seems oddly specific. How was it chosen?

6. I think the authors should consider doing analyses specific per cancer type to convince that there is no confounding. Disease type if not included in the model.

7. The term “genetic”: first, I don't think it's an accurate term, and “genomic” should be used instead. More importantly, I don't think it's specific enough and I often found myself confused when reading. I think “SV” should be used instead (“SV principal component”, “SV profile”, etc.).

8. The claim in L318-321 is unclear. Why is an association between SV principal coordinates and the cohort demonstrated that SV profile difference between cohort is independent to the variation in microbial abundances?

9. Why are only microbial abundances analyzed in meta analysis?

10. Many of the figures are missing a visual legend (e.g., 4A, 5)

11. I think some sort of effect size should somehow be reflected in figure 4A.

12. Was FDR applied across all clinical covariates or per covariate? This should be specified explicitly.

13. Figure 5: It's not clear what are the clusters showing – are these simply the subspecies clusters? Is this the same as Figure S3? Showing the associations themselves would be more interesting here in my opinion.

14. Maybe this is a matter of phrasing, but L441-444 are simply not true. This is not shown by this study.

15. Supplementary figures were very low quality and difficult to evaluate.

Minor comments:

1. L168-183 is really hard to parse. I think the authors should put a list of all clinical abbreviations along with this paragraph. Also, the authors should note that they define CR, PR, SD, PD only after they use them in L173.

2. I think that the table detailing the different cohorts should be a main table.

3. L316 is missing a p-value.

4. I think S1C should be sorted by the strength of the univariate association.

5. In L287-289 the authors mention figure S6 that was not provided (or else they meant S5).

6. L330 – I think should be Figure 4A here.

7. L391-392 – I think the numbers might be swapped. It seems that it is 23 SVs in 16 species.

Reviewer #2 (Remarks to the Author):

The authors have prepared a careful, comprehensive and well-communicated meta-analysis of several studies of the intestinal microbiome with respect to ICI outcomes. Particular strengths include the unsupervised nature of the analysis and the way that significant findings were clearly conveyed. While several associations seen were previously described, this is in a way reassuring regarding the methodology. A particularly novel aspect was the evaluation of SVs and successful identification that several that were significantly associated with outcomes.

One strategy that I don't think they have employed is to see if significant findings from their combined cohort analyses can also be observed in the individual studies. This would be particularly interesting in the multiple melanoma cohorts, but could also be applied to the two NSCLC cohorts. Associations seen trending in the same direction or achieving significance in several individual studies are more compelling, given the evidence for reproducibility.

Central South University

Response to Reviewers

Gut microbial structural variations associate with immune checkpoint inhibitor response

Nature Communications, manuscript ID: NCOMMS-23-19141

Reviewer #1:

Question 1.0. Liu et al. present an analysis of gut microbial abundances and structural variants in the context of response to immune checkpoint inhibitor therapy. They have harmonized a large volume of data (>1K participants) and obtained interesting results. This is an interesting problem and the premise for this work is very compelling. SV analysis could be very interesting in this realm and I think the results, especially at the end of the manuscript + interpretation in the discussion, showcase this potential. I think the manuscript is timely, interesting, largely well written, and the figures are beautiful. Well done to the authors.

Response 1.0:

We feel great thanks for your professional review work on our manuscript. We are pleased that the reviewer considers our study an important and interesting work. As far as you are concerned, there are several problems that need to be addressed. According to your nice suggestions, we have made extensive corrections to our previous draft, and we hope this revised manuscript will satisfy your requirements.

Question 1.1. I unfortunately have significant issues with some of the basic data analysis approaches taken here. I apologize in advance for quite the lengthy review – I really wanted to lay out all the different issues with the intent of helping the authors make this an excellent paper:

Major comments:

1. The authors generally take the approach of analyzing all the studies together. I think this can cause major issues. What I am most concerned about is that SVs are called on all studies together. Since SV “detection” is threshold-ed on passing some level of variability (deletions, for example, have to be deleted in between 25-75% of the samples), this actually biases SGVFinder to find strain differences between the cohorts, rather than to find regions that are truly adaptive within a cohort.

Central South University

Response 1.1: We thank the reviewer for raising the concern.

As you noticed, the current SV detection approach we are using is based on the variability assessment of microbial genomic segments across the samples, and the determination of the SV region border could be different when the sample set is different due to the change in variability value, which made the mapping and replication analysis difficult.

Since the aim of our study is to find replicated associations between SVs of gut microbiota and the host’s response to ICIs treatment, we have followed the replication design of Zeevi D et al. (Zeevi D et al., *Nature*, 2019. PMID: 30918406).[1], which aims to find SVs associated with host health. Zeevi D et al. described the design in the Method part (page 7, **Analysis of replication in Dutch Lifelines DEEP cohort**):” To analyse replication of associations between cohorts, we calculated for each SV region in the Israeli cohort, its presence/absence (deletion SV) or standardized coverage (variable SV) in the Lifelines DEEP cohort. We then tested the association of these regions.....”. Firstly, we check the main information of the cohorts (summarized in the following table). Study cohorts for melanoma (highlighted with yellow) were mainly from the USA (three cohorts), with one cohort from the UK; meanwhile, two studies for NSCLC or RCC were all from France (highlighted with green). Since the associations were conducted per cancer type, we decided to call SVs for studies of melanoma and for studies of NSCLC or RCC separately. We decided to combine the three cohorts from the USA since two of these cohorts were from the University of Texas (FrankelAE_2017 and SpencerCN_2021). The cohorts used to call SVs, which are utilized as references (with a larger sample size), were listed in the last column of the following table. This process tries to avoid the confounding brought up by race and make replication easier.

Study	N	Cancer type included	Sample source	Call SV
Frankel et al., Neoplasia 2017	39	Melanoma	University of Texas Southwestern Medical Center	Joint to call
McCulloch et al., Nature Medicine. 2022	62	Melanoma	University of Pittsburgh’s Hillman Cancer Center (HCC)	SV_USA

Central South University

Spencer et al., Science 2021	167	Melanoma	University of Texas MD Anderson Cancer Center (UTMDACC) in Houston	
Lee et al. Nature Medicine. 2022	164	Melanoma	United Kingdom	Call SVs with SV_USA as reference
Derosa et al. Nature Medicine. 2022	338	Non-small-cell lung cancer	12 academic centers in France and 2 in Canada	call SV_France
Routy et al., Science 2018	219	Non-small-cell lung cancer, Renal cell carcinoma	France	Call SVs with SV_France as reference

The association analysis was conducted for different cancer types. As for melanoma, we detected SVs in three cohorts from the USA (FrankelAE_2017, McCullochJA_2022, and SpencerCN_2021, 268 samples) using SGV-Finder with default parameters (except --min_samp_cutoff 27). To analyze the replication of associations between cohorts for melanoma, we calculated for each SV region in the USA cohort its dSV or vSV in the LeeKA_2022 cohort from the United Kingdom. We run SGV-Finder with the -by-orig parameter by using the orig_dsgv.df and orig_vsgv.df files generated with the combined USA cohort. Meanwhile, SVs were identified across 338 samples from France within the DerosaL_2022 dataset using ICRA and SGV-Finder with default parameters (except -min_samp_cutoff 34). After that, we calculated for each SV region in the DerosaL_2022, its dSV or vSV in the RoutyB_2018 cohort, which is also from France, and ran SGV-Finder with the -by-orig parameter by using the orig_dsgv.df and orig_vsgv.df files generated with the DerosaL_2022 cohort.

In the SV calling process, the -min_samp_cutoff parameter was determined to be about 10% of the total sample size (the default number is 75, which is too strict for our relatively small sample size cohort). In the publication of Zeevi D et al. (Zeevi D et al., Nature, 2019. PMID: 30918406), the -min_samp_cutoff was set to be 75, which is about 10% of the total samples

Central South University

(880) used for the SV call.

With the above process, we unified the SV borders for melanoma and NSCLC or RCC, followed by the replication association analysis.

We agree with the reviewer that the joint-calling approach may miss the population-specific SVs that are variable only in one cohort and failed to pass the deletion proportion cut-off in the merged population. However, in this study, we focus on replicable associations, meaning that the associated SVs should be present in at least two cohorts, so the absence of a population-specific association doesn't affect the downstream analysis. What's more, we adjust the association analysis, which was conducted within each cohort, for each clinical outcome per cancer type. After that, we combine the results with meta-analysis to find replicated associations.

Question 1.2.

a. How many of the SVs have a significant difference between the studies?

Response 1.2: Thanks for this question.

In total, for studies conducted for melanoma, we detected 7,879 SVs (5,595 dSVs and 2,284 vSVs) by joint SV calling in the USA dataset, and 2298 (29.16%) SVs showed significant differences between four studies (three USA and one from the UK) (normal $P < 0.05$), including 779 (34.1%) vSVs and 1519 (27.14%) dSVs.

Meanwhile, for studies conducted for NSCLC or RCC, we detected 6,427 SVs (2,053 vSVs and 4,374 dSVs) by SV calling in the France dataset, and 637 (9.91%) SVs showed significant differences between two studies (normal $P < 0.05$), including 316 (15.3%) vSVs and 321 (7.33%) dSVs.

The difference between studies for dSVs was tested with a chi-square test, which was performed with the R `chisq.test` function, and the difference between studies for vSVs was tested with a nonparametric Kruskal-Wallis test, which was performed with the R `kruskal_test` function.

Question 1.3.

b. This is probably my most important comment on this paper: Since this issue involves feature construction and not just data analysis, I strongly believe that just including the study in the

Central South University

regression model is not going to be enough. I think the best analytic approach would be to identify SVs within each study and then check for generalization (i.e., take L392-393 and the supp tables, and make them the focus of the manuscript), but I understand that this loses power significantly. In lieu of that, I think the authors should do a cross-study validation – i.e., detect SVs and their associations within 6 of the studies, and then check whether the associations replicate within the same study when SVs in that study are called with SGVFinder’s “by_orig” parameter. Then repeat this procedure for all studies. Then do downstream analyses only on SVs that truly generalize across studies.

c. The fact that the authors find varying enrichment in different cohorts (S4, L373-375) strongly supports this concern.

Response 1.3: Thanks for your nice comments on our manuscript. Firstly, we add a flowchart of the study that details the samples utilized at each stage of statistical analysis. (Figure S1).

Central South University

Figure S1. A flowchart of the study that details the samples utilized at each stage of statistical analysis. OS, Overall survival. OS: overall survival; PFS: progression-free survival; irAEs: immune-related adverse events. NSCLC: Non-small-cell lung cancer; RCC: Renal cell carcinoma.

Central South University

The sample size and cohort for each clinical outcome of different cancer type were listed in following table.

Cancer type	Clinical outcome	Cohorts (N*)	Total sample size	Country	Candidate association SV selection criteria
Melanoma	response	FrankelAE_2017 (39); LeeKA_2022 (162); McCullochJA_2022 (62); SpencerCN_2021 (155)	418	USA/UK	(1) $P_{meta} \leq 0.01$; (2) $P_{heterogeneity} > 0.05$; (3) Normal $P \leq 0.2$ within at least two cohorts and the trending in the same direction.
	PFS12	LeeKA_2022 (162); McCullochJA_2022 (62); SpencerCN_2021(109)	333	USA/UK	
	OS	McCullochJA_2022 (62)	62	USA	FDR p-value ≤ 0.1
	irAEs	McCullochJA_2022 (62)	62	USA	
NSCLC	response	DerosaL_2022 (336); RoutyB_2018 (118)	454	12 academic centers in France and 2 in Canada	(1) $P_{meta} \leq 0.01$; (2) $P_{heterogeneity} > 0.05$; (3) Normal $P < =0.2$ within at least two cohorts and the trending in the same direction.
	OS	DerosaL_2022 (336)	336	12 academic centers in France and 2 in Canada	FDR p-value ≤ 0.1
RCC	response	RoutyB_2018 (101)	101	France	FDR p-value ≤ 0.1

We did an association analysis for each cohort for different clinical outcomes per cancer type. Although the total sample size is 973, the sample size is small for some clinical outcomes; for example, only 62 samples from one cohort can be utilized to test the association between irAEs and SVs. Anywhere, almost all publicly available immune checkpoint inhibitor studies

Central South University

with metagenome data all over the world were collected by us. As we listed in Table S1A, some datasets need requests, and we have failed to get permission to obtain the datasets after sending an email to the corresponding PIs. The limitation of a relatively small sample size is also added in the revised manuscript of the discussion part (page 20, line 561-564): "The sample size for association tests of clinical outcomes, such as OS and irAEs, per cancer type is small, which results in a lack of statistical power. The correlations between prognosis after ICIs therapy and microbial SVs call for validation in additional populations with a larger sample size."

We have defined different criteria for selecting candidate associated SVs when there is only one dataset or at least two datasets exist (listed in the last column of the above table). We focus on finding replicated associations between different cohorts. If there is only one dataset for clinical outcome, we considered only the significant SV-prognosis associations with an FDR p-value < 0.1 . If there is more than one dataset for clinical outcome, the replicated candidate associations were confirmed with the following three criteria: (1) $P_{\text{meta}} \leq 0.01$; (2) $P_{\text{heterogeneity}} > 0.05$; and (3) $P_{\text{normal}} \leq 0.2$ within at least two cohorts and trending in the same direction. We know this criteria is loose; we use the above criteria to find candidate associations. We avoid using the expression "statistically significant", and use candidate associated". FDR adjustment design to control the false positive rate, but it is strict and may cause false negatives. We hope our study can act as resources and give some clues for experiment validation rather than just focusing on the statistical significance. Associations for every clinical outcome per cancer type with a meta-p value ≤ 0.05 when there are more than one cohort or a normal p-value ≤ 0.05 when there is only one dataset exist were all listed in Supplementary Table 5. The annotations for each SV were also provided.

We have updated the whole manuscript with new analysis results, as mentioned above. Please check it out. If you have any suggestions on how to select candidate-associated SVs, please tell me.

Question 1.4

2. Association analysis:

a. Why would the authors include read count as a confounder? It seems that by construction SVs should not be affected by read depth, and the analysis in Fig. S1c actually supports that.

Response 1.4: Thanks for pointing this out. Yes, theoretically, the SV profile shouldn't be

Central South University

affected by the read count since the coverage across different genomes and different samples has been standardized. We have excluded read counts from the association models in the revised manuscript and updated all the results. Please check it out. Thanks!

Question 1.5

b. Why would the authors include species relative abundance in the model? That also seems unrelated for largely the same reasons. Is there an association between SVs of a microbe and the relative abundances of the same microbe? (not this is not the same as the association with microbiome PCs shown in Fig. S1C, unless something is very unclear in the description of the regression model).

Response 1.5: Thanks for your suggestions. As reported in former publications (Routy et al. Science. 2018.PMID: 29097494; Derosa et al. Nature medicine. 2022. PMID: 35115705; Gopalakrishnan et al. Science. 2018. PMID: 29097493. Matson et al. Science. 2018. PMID: 29302014; Frankel et al. Neoplasia. 2017. PMID: 28923537; Lee et al. Nature medicine. 2022. PMID: 35228751) [2-7]. There are associations between the species abundance and response to ICIs of patients, and our meta-analysis has confirmed this. As you can see in Figures 3A and 3B, the purple and yellow boxes indicate the association between clinical outcomes and species abundance. So that when testing the association between SV and clinical outcomes after ICIs treatment, to remove the confounding effect of species abundance and identify the pure SV level associations, just as the team of Jing Yan Fu et al. did in their publications (Wang D, et al. Cell Host Microbe. 2021. PMID: 34847370; Chen L, et al. Cell. 2021. PMID: 33838112; Chen L, et al. Nature medicine. 2022. PMID: 36216932)[8-10]. We aim to find an independent association between SV and clinical outcomes after ICI treatment that is independent of microbial abundance, so that the corresponding species abundance is included as a covariate in the model.

Central South University

Question 1.6 3. Distance calculations:
 a. I find it very odd to combine together continuous features (vSVs) and binary features (dSVs). Perhaps the authors should calculate one distance for each and average them.

Response 1.6: We think this is an excellent suggestion. In the revised manuscript, we measured the variability of microbial vSVs between samples using the Canberra distance metric, calculated the variability of microbial dSVs between samples using the Jaccard distance metric, and then measured the variability of microbial SVs between samples using the average value of the above two distance matrixes. As Rob Knight et al. recommended in the review, (Knight R, et al. Nat Rev Microbiol. 2018. PMID: 29795328) [11] (page 417, **Higher- level analyses** section):” Quantitative metrics (Bray–Curtis, Canberra and weighted UniFrac) use feature abundance data in calculations, whereas qualitative metrics (binary-Jaccard and unweighted UniFrac) only consider the presence or absence of features”. We have states this in the

Central South University

method of the revised manuscript. The corresponding text now reads:

Page 9-10, line 254-259:

"In this study, we measured the variability of microbial vSVs between samples using the Canberra distance metric, and calculate the variability of microbial dSVs between samples using the jaccard distance metric, then measure the variability of microbial SVs between samples using the average value of above two distance matrix. We accomplished this by computing the distance of SVs utilizing the vegdist() function from the R package vegan (version 2.5.6)."

Question 1.7. b. The authors should justify or give intuition for the use of the Canberra distance.

Response 1.7: The reason we use Canberra distance is according to the recommendation of Rob Knight et al. (Knight R, et al. Nat Rev Microbiol. 2018. PMID: 29795328) [11] (page 417, **Higher- level analyses** part):" Quantitative metrics (Bray–Curtis, Canberra and weighted UniFrac) use feature abundance data in calculations, whereas qualitative metrics (binary-Jaccard and unweighted UniFrac) only consider the presence or absence of features". We have added this reference in the revised manuscript (page 9, line 256).

Question 1.8. c. What does "distance of shared SVs" (L240) mean? Is there a different dimensionality per sample pair? That could cause substantial issues in data analysis. If not – how are missing data handled?

Response 1.8: Thanks for pointing this out. Firstly, we didn't do any imputation for missing values. There is no different dimensionality per sample pair. The shared SVs mean that we calculate the distance of the sample based on the SVs that both exist (and are not missing) in the two samples. In case it may cause misunderstandings, we delete the "shared" word. The missing data is not included in the whole data analysis. They are deleted. For datasets like the following:

Central South University

SVs	Sample A	Sample B
dSV1	1	0
dSV2	0	NA
vSV3	0.7	3.4
vSV4	NA	2.4
dSV5	1	1

We will use the following data matrix for distance calculation:

SVs	Sample A	Sample B
dSV1	1	0
vSV3	0.7	3.4
dSV5	1	1

Further, we conducted association analysis after deleting the missing data, and association analysis was just conducted with a complete data matrix. We provide the sample size after deleting the missing data when conducting association analysis for each SV in Supplementary File 3 and Supplementary File 4.

Question 1.9. 4. Clustering analysis within species:

a. I don't understand the premise of this analysis. Is this supposed to be capturing sub-species information? Wouldn't mOTUs2 or MetaPhlan4 be more suited for this purpose? The authors should benchmark their explanatory power if this is their intent. (Running Kraken on Progenomes1 wouldn't cut it here).
I should note that I think this analysis misses the point of SVs which is to narrow down to a specific genomic region.

Response 1.9: Thanks for your nice suggestions. We agree with your suggestion and delete

Central South University

the clustering analysis with species section in the revised manuscript. Further, we focus on the analysis point of SVs, which is to narrow down to a specific genomic region. We adjust the association method and provide more compressive results (Figure 5, Figure 6, and Table S4).

Question 1.10. b. The associations in Fig. 4B-G seem extremely weak. Is this even significant? Was this adjusted for multiple testing? P-values are not shown anywhere (or written in the text).

Response 1.10: We thank the reviewer for this comment. The analysis was conducted with SVs, and the P-values from the Adonis test were added to the picture. An example picture is shown as follows:

The p-value is not adjusted for multiple testing; it's the association between the SV matrix and clinical outcomes when adjusting for species abundance, age, and gender. We conducted the associations in each cohort and combined the results with meta-analysis. The ones with $P_{meta} < 0.05$ and $P_{heterogeneity} > 0.05$ were considered significant (Figures 4A, 4B, and Table S3). Figures 4C–4F are examples of significant associated SV species within one cohort.

Question 1.11. c. Is the model written out in L268 applied in the permanova somehow, or is

Central South University

this unadjusted? It's very problematic if unadjusted.

Response 1.11: We thank the reviewer for this comment. When analyzing the association between species SV as a whole and clinical outcomes (binary, such as response to ICI drugs, irAEs, or whether PFs were longer than 12 months), permanova were conducted, with the distance matrix of SV within a species as the dependent variable, clinical outcomes as the independent variable, and age and gender also included as covariates. Meanwhile, we conduct association studies for SVs; at this point, we have different models based on the character of the clinical outcomes. Just as Zeevi D et. al (Zeevi D, et al. Nature. 2019. PMID: 30918406)[1] describe in the method (page 8, **Association of SVs with disease risk factors**):" Variable SVs were correlated (Spearman's) to disease risk factors (MAP, waist circumference, and so on) and P values were calculated, ensuring a minimum of 20 subjects in each comparison. Two-sided Mann–Whitney U-test was used to calculate significance of associations between deletion SVs and the same disease risk factors, demanding at least 5 subjects in each comparison and at least 5 unique values in each group." Also the publication of Wang et. al, they use linear regression with dependent variable being bile acid metabolism index, and independent variable be SVs. To test the associations between SVs and clinical outcome, we use logistic regression models for binary variables and cox-regression models for survival data. Since the sample size of our study is relatively small, we use different criteria for the selection of candidate association SVs, as described in response 1.3:

Cancer type	Clinical outcome	Cohorts (N*)	Total sample size	Country	Candidate association SV selection criteria
Melanoma	response	FrankelAE_2017 (39); LeeKA_2022 (162); McCullochJA_2022 (62); SpencerCN_2021 (155)	418	USA/UK	(1) $P_{meta} \leq 0.01$; (2) $P_{heterogeneity} > 0.05$; (3) Normal $P \leq 0.2$ within at least two cohorts and the trending in the same direction.
	PFS12	LeeKA_2022 (162); McCullochJA_2022 (62); SpencerCN_2021(109)	333	USA/UK	

Central South University

	OS	McCullochJA_2022 (62)	62	USA	FDR p-value ≤ 0.1
	irAEs	McCullochJA_2022 (62)	62	USA	
NSCL C	response	DerosaL_2022 (336); RoutyB_2018 (118)	454	12 academic centers in France and 2 in Canada	(1) $P_{meta} \leq 0.01$; (2) $P_{heterogeneity} > 0.05$; (3) Normal $P \leq 0.2$ within at least two cohorts and the trending in the same direction.
	OS	DerosaL_2022 (336)	336	12 academic centers in France and 2 in Canada	FDR p-value ≤ 0.1
RCC	response	RoutyB_2018 (101)	101	France	FDR p-value ≤ 0.1

We have re-write this part of method as follows in the revised manuscript.

Page 10 - 11, line 267 – 306:

“Association analysis

We examined the differences in response, PFS12, and irAEs following ICI treatment across the SV or abundance of species. The associations between OS and SV, or abundance, for each species were also investigated. The assignment values of the clinical variables were listed in Table S1C.

i) Species-level associations of the gut microbiome with clinical outcomes

The association between binary clinical outcomes and SV of each species was evaluated using PERMANOVA with 999 permutations with the following formula:

distance matrix of SV ~ clinical outcome + Species relative abundance + Age_bins + Gender

The association between binary clinical outcomes and species relative abundance was

Central South University

evaluated using a logistic regression model with the following formula:

$$\text{clinical outcome} \sim \text{Species relative abundance} + \text{Age_bins} + \text{Gender}$$

The relationship between species relative abundance and OS were using cox regression model with formula:

$$\text{OS} \sim \text{Species relative abundance} + \text{Age_bins} + \text{Gender}$$

For each clinical outcomes, association analysis were performed within each cohort, and meta-analysis with a random-effect model was performed to integrate the results of different cohorts.

ii) dSV or vSV site based associations of the gut microbiome with clinical outcomes

Associations between SVs and binary clinical outcomes were assessed using logistic models with the formula:

$$\text{clinical outcome} \sim \text{SV} + \text{Age_bins} + \text{Gender} + \text{Species relative abundance}$$

, demanding at least 10 subjects in each comparison and at least 3 responders.

Associations between SVs and OS were assessed using Cox regression models with the formula:

$$\text{OS} \sim \text{SV} + \text{Age_bins} + \text{Gender} + \text{Species relative abundance}$$

,ensuring a minimum of 20 subjects in each comparison.

In order to derive more easily interpretable HRs or ORs, quartiles (25%, 50%, and 75%) of the value of each vSV were computed for association analysis and modeled as continuous variables.

The Benjamini-Hochberg (false discovery rate: FDR) P-value correction method was applied with the *p.adjust()* function in R. Specifically, for vSVs, dSVs, and species relative abundance, we carried out association analysis and P-value correction independently. If there is only one dataset for clinical outcome, we considered the SV-prognosis candidate associations with an FDR p-value ≤ 0.1 . If there is more than one dataset for clinical outcome, the replicated candidate associations were confirmed with the following three criteria: (1) $P_{\text{meta}} \leq 0.01$; (2) $P_{\text{heterogeneity}} > 0.05$; (3) Normal $P \leq 0.2$ within at least two cohorts and trending in the same direction (Table S1D).

We calculated the Spearman's correlation coefficient between effect sizes within different cancer types for the analysis presented in Figs. S5A–S5D.”

Central South University

Question 1.12. Other comments:

1. The authors mention that they preferred dSVs to vSVs – why? I think both should be kept, and in any case, vSVs should have more resolution.

Response 1.12: Thanks for reminding us of this. We didn't prefer dSVs to vSVs. I think this impression may be caused by the fact that dSV was mentioned more than vSV in the results and discussion parts. We have corrected this. Please check Figure 5, Figure 6, and the results section in the revised manuscript.

Question 1.13. 2. The sentences in L243-247 are unclear. What do the authors mean by "PCs [...] were chosen"? What was done with them? Later in the discussion (L451-453) they seem to imply that you didn't do anything with them?

Response 1.13: In the revised manuscript, we have deleted the sentences "To address the possibility that genetic differences between different cohorts might be influenced by species composition, the top five principal components (PCs) of a microbial abundance profile that explained over 60% of the overall compositional variance were chosen. Thanks!

Question 1.14. 3. How was missing data handled in general in the analysis? Did the authors only compare dSVs / vSVs within samples in which the species was identified?

Response 1.14: Thanks for this comment. As we responded in Question 1.8.

Firstly, we didn't do any imputation for missing values. There is no different dimensionality per sample pair. The missing data is not included in the whole data analysis. They are deleted. For datasets like the following:

SVs	Sample A	Sample B
dSV1	1	0
dSV2	0	NA
vSV3	0.7	3.4

Central South University

vSV4	NA	2.4
dSV5	1	1

We will use the following data matrix for distance calculation:

SVs	Sample A	Sample B
dSV1	1	0
vSV3	0.7	3.4
dSV5	1	1

Further, we conducted association analysis before deleting the missing data, and association analysis was just conducted with a complete data matrix. We provide the sample size after deleting the missing data when conducting association analysis for each SV in Supplementary File 3 and Supplementary File 4. For example, in Supplementary File 4A, for SV "[Clostridium] leptum DSM 753:3270_3271" (line #1), In cohort FrankelAE_2017, there are 13 samples left for association analysis when deleting the samples with any of the values of [Clostridium] leptum DSM 753:3270_3271, response, age, and gender missing. We only compare dSVs and vSVs within samples in which the species was identified.

Question 1.15. 4. The authors should specify the parameters for running SGVFinder.

"min_samp_cutoff" is of particular interest here.

Response 1.15: Thanks for pointing this out. The min_samp_cutoff is indeed an important parameter for SV calls. The min_samp_cutoff was determined as about 10% of the total sample size (the default number is 75, which is too strict for our relatively small sample size cohort; we calculate that 75/880 is about 10% in the publication of Zeevi D et al. (Zeevi D, et al., Nature, 2019. PMID: 30918406). We added the -min_samp_cutoff parameter in the revised manuscript. The corresponding text now reads:

Page 8, line 217-220:

"SVs were detected based on the high-quality metagenomic sequence reads. Overall, we detected

Central South University

SVs across 268 samples (FrankelAE_2017, McCullochJA_2022, and SpencerCN_2021) from the USA using ICRA and SGV-Finder with default parameters (except --min_samp_cutoff 27)."

page 8, lines 224-226: "Moreover, SVs detected across 338 samples (Derosal_2022) from France were identified using SGV-Finder with default parameters (except --min_samp_cutoff 34)."

Question 1.16. 5. L259: A cutoff of 0.55 seems oddly specific. How was it chosen?

Response 1.16: Thanks for this comment. Tibshirani and Walther (2005) recommend choosing as the optimal number of clusters the largest number of clusters that leads to a prediction strength above 0.8 or 0.9 (<http://127.0.0.1:18654/library/fpc/html/prediction.strength.html>). The whole clustering analysis was deleted in the revised manuscript, as you suggested in Question 1.9.

Question 1.17. 6. I think the authors should consider doing analyses specific per cancer type to convince that there is no confounding. Disease type if not included in the model.

Response 1.17: Thanks for your suggestion. As described in the following table, we did analyses for different clinical outcomes per specific cancer type. So different models were established in each cohort for each clinical outcome per cancer type. For example, when testing the association between SV and response to ICIs within melanoma, we developed four models within cohort FrankelAE_2017 (with 39 samples); LeeKA_2022 (with 162 samples); McCullochJA_2022 (with 62 samples); and SpencerCN_2021 (with 155 samples), then combined the results of the above four cohorts with meta-analysis. We repeat this process for each clinical outcome per cancer type. So it is not necessary to include disease type in the model since in each model, the value of disease type is the same (melanoma, NSCLC, or RCC).

Cancer type	Clinical outcome	Cohorts (N*)	Total sample size	Country
Melanoma	response	FrankelAE_2017 (39); LeeKA_2022 (162); McCullochJA_2022 (62); SpencerCN_2021 (155)	418	USA/UK

Central South University

	PFS12	LeeKA_2022 (162); McCullochJA_2022 (62); SpencerCN_2021(109)	333	USA/UK
	OS	McCullochJA_2022 (62)	62	USA
	irAEs	McCullochJA_2022 (62)	62	USA
NSCLC	response	DerosaL_2022 (336); RoutyB_2018 (118)	454	France
	OS	DerosaL_2022 (336)	336	France
RCC	response	RoutyB_2018 (101)	101	France/Canda

Question 1.18. 7. The term “genetic”: first, I don’t think it’s an accurate term, and “genomic” should be used instead. More importantly, I don’t think it’s specific enough and I often found myself confused when reading. I think “SV” should be used instead (“SV principal component”, “SV profile”, etc.).

Response 1.18: We thank the reviewer for this comment. We have replaced "genetic" with "SV" in the whole revised manuscript.

Question 1.19. 8. The claim in L318-321 is unclear. Why is an association between SV principal coordinates and the cohort demonstrated that SV profile difference between cohort is independent to the variation in microbial abundances?

Response 1.19: We thank the reviewer for this comment. We have modified this sentence as follows in the revised manuscript. The corresponding text now reads:

Page 13, lines 344-348:

“ Further, the genetic principal coordinates (PCo) 1 and PCo2 demonstrated differences between these four cohorts (ANOVA test, $P = 4.32e-11$ for PCo1 and $P < 2e-16$ for PCo2), demonstrating a divergence of microbial SVs between these four cohorts that was independent of differences in their microbial abundances.”

Question 1.20. 9. Why are only microbial abundances analyzed in meta analysis?

Central South University

Response 1.20: We thank the reviewer for this comment. We did meta-analyses for both the microbial abundances and SVs of species. The results can be found in Supplementary Table S3. For example, in table S3A The sv.meta.p column is the meta-analysis p value, which was calculated with the result of cohort McCullochJA_2022 (McCullochJA_2022.sv.P, McCullochJA_2022.sv.N), FrankelAE_2017 (FrankelAE_2017.sv.P, FrankelAE_2017.sv.N), SpencerCN_2021 (SpencerCN_2021.sv.P, SpencerCN_2021.sv.N), and LeeKA_20221 (LeeKA_20221.P, LeeKA_20221.sv.N). We have modified the expression in the method part of the revised manuscript. The corresponding text now reads:

Page 9, lines 249-250:

"Meta-analysis with a random-effect model was performed to integrate the results of different cohorts."

Question 1.21. 10. Many of the figures are missing a visual legend (e.g., 4A, 5)

Response 1.21: We thank the reviewer for this comment. We have checked for every figure and supplementary file in the manuscript and made sure that every figure has a visual legend. Figures 4A and 4B have a legend as follows:

Central South University

Question 1.22. 11. I think some sort of effect size should somehow be reflected in figure 4A.

Response 1.22: We thank the reviewer for this comment. We have modified the sorting of the species by effect size (the sum number of associated clinical outcomes) in Figures 4A and 4B. For example, for species *D. formicigenerans*, its abundance and SV are associated with response, and for PFS12 of melanoma, its SV is associated with irAEs. So its associated number is 5. We sort species by this number.

Question 1.23. 12. Was FDR applied across all clinical covariates or per covariate? This should be specified explicitly.

Response 1.23: Thanks for this comment. FDR was applied per covariate within different clinical outcomes of different cancer types, also for dSV and vSV, respectively. We have specified explicitly in the revised manuscript. The corresponding text now reads:

Page 11, line 297-300:

"The Benjamini-Hochberg (false discovery rate: FDR) P-value correction method was applied with the p.adjust() function in R. Specifically, for vSVs, dSVs, and species relative abundance, we carried

Central South University

out association analysis and P-value correction independently."

Question 1.24. 13. Figure 5: It's not clear what are the clusters showing – are these simply the subspecies clusters? Is this the same as Figure S3? Showing the associations themselves would be more interesting here in my opinion.

Response 1.24: We thank the reviewer for this comment. Since the clustering analysis was deleted in the revised manuscript, Figure 5 is removed accordingly.

Question 1.25. 14. Maybe this is a matter of phrasing, but L441-444 are simply not true. This is not shown by this study.

Response 1.25: Thanks for pointing this out. We have replaced this sentences with “Notably, this is the first investigation into the microbial genetic determinants of response to ICIs in humans. We provide some clues for further mechanistic studies to explore how gut microbiota modulate antitumor immunity and affect the efficacy of ICIs.”(page 18, lines 492 - 495) in the revised manuscript.

Question 1.26. 15. Supplementary figures were very low quality and difficult to evaluate.

Response 1.26: Thanks for reminding us. We have modified the supplementary files to make them of higher quality and easier to evaluate. Those figures corresponding to clustering analysis and of low quality were removed from the revised manuscript.

Question 1.27. Minor comments:

1. L168-183 is really hard to parse. I think the authors should put a list of all clinical abbreviations along with this paragraph. Also, the authors should note that they define CR, PR, SD, PD only after they use them in L173.

Response 1.27: We thank the reviewer for this comment. We have added a list of all clinical abbreviations along with the paragraph. Further, we have re-written this paragraph to make it easy to understand as follows in the revised manuscript. The corresponding text now reads:

Page 7 lines 170-188:

Central South University

Definition of clinical outcomes

The study collected clinical information such as age, gender, ICI targets, PFS, and OS. While recognizing that the current definition of response is conservative and that patients who have stable disease (SD) and have extended survival can be thought of as experiencing clinical benefit from ICI treatment, we employed the following definition to ensure consistency with recent literature and clear response interpretation[12, 13], which was determined using Response Evaluation Criteria in Solid Tumors (RECIST) criteria for radiological response as represented in the original articles. Responders were determined as patients who demonstrated a complete response (CR) or partial response (PR), while non-responders had SD or progressing disease (PD). The definition of progression-free survival at 12 months (PFS12) was that there was no disease progress as evaluated by RECIST 12 months after ICI's treatment initiation. Response, PFS12, OS, and irAEs were utilized as clinical outcomes to ensure strict consistency in outcome measurement across the six studies (Table 1).

The abbreviations for clinical outcomes are listed as follows: PFS: progression-free survival; OS: overall survival; SD: stable disease; CR: complete response; PR: partial response; PD: progressing disease; PFS12: progression-free survival at 12 months; irAEs: immune-related adverse events."

Question 1.28. Minor comments:

2. I think that the table detailing the different cohorts should be a main table.

Response 1.28: I think it is a nice suggestion. We have added Table 1 (just as follows) detailing the main characteristics of the six different cohorts in the revised manuscript.

Table 1. Summary of ICIs metagenome studies.

Study	Study name utilized in this study	Accession number	N*	Cancer type included	Treatment	Clinical outcomes	Country
Frankel et al., Neoplasia 2017	FrankelAE_2017	PRJNA397906	39	Melanoma	CTLA4/PD-1 blockade	Response	USA
McCulloch et al., Nature Medicine. 2022	McCullochJA_2022	PRJNA762360	62	Melanoma	PD-1 blockade	Response/irAEs /OS/PFS	USA
Spencer et al., Science 2021	SpencerCN_2021	PRJNA770295	167	Melanoma	PD-1 blockade	Response/PFS	USA

Central South University

Lee et al.Nature Medicine.2022	LeeKA_2022	PRJEB431 19	164	Melanoma	CTLA4/PD-1 blockade	Response/PFS_ 12	UK
Routy et al., Science 2018	RoutyB_2018	PRJEB228 63	219	NSCLC; RCC	PD-1 blockade	Response	France
Derosa et al.Nature Medicine.2022	DerosaL_2022	PRJNA751 792	338	NSCLC	PD-1 blockade	Response/OS	France/Canda

Abbreviations: OS: overall survival; PFS: progression-free survival; NSCLC: Non-small-cell lung cancer; RCC: Renal cell carcinoma; PFS_12: whether PFS longer than 12 months; irAEs: immune-related adverse events.

* Samples collected pretreated with ICIs and with matched clinical and metagenome data available.

Question 1.29. 4. I think S1C should be sorted by the strength of the univariate association.

Response 1.29: We thank the reviewer for this comment. The S2C and S3C in the revised manuscript have been sorted by the strength of the univariate association just as follows:

Please check it out.

Question 1.30. 5. In L287-289 the authors mention figure S6 that was not provided (or else they meant S5).

Response 1.30: I'm sorry for this careless mistake. We have corrected this in the manuscript.

Please check it out.

Question 1.31. 6. L330 – I think should be Figure 4A here.

Central South University

Response 1.31: I'm sorry for this careless mistake. We correct this in the revised manuscript.

Thanks!

Question 1.32. 7. L391-392 – I think the numbers might be swapped. It seems that it is 23 SVs in 16 species.

Response 1.32: Thanks for reminding us. We corrected this in the revised manuscript. Thanks!

Central South University

Reviewer #2:

Question 2.0. The authors have prepared a careful, comprehensive and well-communicated meta-analysis of several studies of the intestinal microbiome with respect to ICI outcomes. Particular strengths include the unsupervised nature of the analysis and the way that significant findings were clearly conveyed. While several associations seen were previously described, this is in a way reassuring regarding the methodology. A particularly novel aspect was the evaluation of SVs and successful identification that several that were significantly associated with outcomes.

One strategy that I don't think they have employed is to see if significant findings from their combined cohort analyses can also be observed in the individual studies. This would be particularly interesting in the multiple melanoma cohorts, but could also be applied to the two NSCLC cohorts. Associations seen trending in the same direction or achieving significance in several individual studies are more compelling, given the evidence for reproducibility.

Response 2.0:

Thanks for your professional and rigorous revision of this manuscript, as well as the fair and constructive comments! We are pleased that the reviewer considers our study an important and interesting work. Thanks for helping us improve the quality of this manuscript.

We have modified the association analysis process according to your and review #1's suggestions.

Firstly, we add a flowchart of the study that details the samples utilized at each stage of statistical analysis. (Figure S1). The sample size and cohort for each clinical outcome of a different cancer type were listed in the following table S1C.

Central South University

Central South University

Figure S1. A flowchart of the study that details the samples utilized at each stage of statistical analysis. OS, Overall survival. OS: overall survival; PFS: progression-free survival; irAEs: immune-related adverse events. NSCLC: Non-small-cell lung cancer; RCC: Renal cell carcinoma. PFS12: progression-free survival at 12 months .

Central South University

Cancer type	Clinical outcome	Cohorts (N*)	Total sample size	Country	Candidate association SV selection criteria
Melanoma	response	FrankelAE_2017 (39); LeeKA_2022 (162); McCullochJA_2022 (62); SpencerCN_2021 (155)	418	USA/UK	(1) $P_{meta} \leq 0.01$; (2) $P_{heterogeneity} > 0.05$; (3) Normal $P \leq 0.2$ within at least two cohorts and the trending in the same direction. FDR p-value ≤ 0.1
	PFS12	LeeKA_2022 (162); McCullochJA_2022 (62); SpencerCN_2021(109)	333	USA/UK	
	OS	McCullochJA_2022 (62)	62	USA	
	irAEs	McCullochJA_2022 (62)	62	USA	
NSCLC	response	DerosaL_2022 (336); RoutyB_2018 (118)	454	12 academic centers in France and 2 in Canada	(1) $P_{meta} \leq 0.01$; (2) $P_{heterogeneity} > 0.05$; (3) Normal $P \leq 0.2$ within at least two cohorts and the trending in the same direction. FDR p-value ≤ 0.1
	OS	DerosaL_2022 (336)	336	12 academic centers in France and 2 in Canada	
RCC	response	RoutyB_2018 (101)	101	France	FDR p-value ≤ 0.1

We take association analysis for each cohort for different clinical outcomes of different cancer types. The sample size is small and leads to small statistical power, and almost all cohorts all over the world were collected by us. Some datasets need requests, and we have tried to send an email to the corresponding PIs to ask for data sharing but failed. We also tried to collect our samples, but it requires time. The limitation of a relatively small sample size is

Central South University

also added in the revised manuscript of the discussion part (page 20, lines 561-564): "The sample size for association tests of clinical outcomes, such as OS and irAEs, per cancer type is small, which results in a lack of statistical power. The correlations between prognosis after ICIs therapy and microbial SVs call for validation in additional populations with a larger sample size."

When analyzing the association between species SV as a whole and clinical outcomes (binary, such as response to ICI drugs, irAEs, or whether PFs were longer than 12 months), permanova were conducted, with the distance matrix of SV within a species as the dependent variable, clinical outcomes as the independent variable, and age and gender also included as covariates. Meanwhile, we conduct association studies for SVs; at this point, we have different models based on the character of the clinical outcomes. We use logistic regression models for binary clinical outcomes and cox-regression models for survival data. Since the sample size of our study is relatively small, we use different criteria for the selection of candidate association SVs, as described in the table above.

We hope we can use our data for validation in future studies. We define different criteria when there is only one dataset or more than two datasets listed in the above dataset. We focus on whether the direction and associations can be replicated in different cohorts. If there is only one dataset for clinical outcome, we considered only the significant SV-prognosis associations with an FDR p-value ≤ 0.1 . If there is more than one dataset for a clinical outcome, the replicated candidate associations were confirmed with the following three criteria: (1) $P_{meta} \leq 0.01$; (2) $P_{heterogeneity} > 0.05$; (3) Normal $P \leq 0.2$ within at least two cohorts and trending in the same direction. We know this is a relatively loose criteria; we found there is no SV that can pass the FDR adjustment of the meta p-value when there are more than two datasets. However, strict FDRA adjustment may cause false negatives. We hope our study can act as a resource for reference and give some clues for experiment validation rather than just focusing on the statistical significance. Associations with a meta-p value < 0.05 when there are more than one cohort or a normal p-value < 0.05 when only one dataset exists were all listed in Supplementary Table 5. The annotations for each SV were also provided.

The updated results can be found in supplementary tables 3 and 4. We have updated the whole manuscript with new analysis results, as mentioned above. Please check it out. Thanks!

1. Zeevi, D., et al., *Structural variation in the gut microbiome associates with host health*. Nature, 2019. **568**(7750): p. 43-48.

Central South University

2. Routy, B., et al., *Gut microbiome influences efficacy of PD-1-based immunotherapy against epithelial tumors*. Science, 2018. **359**(6371): p. 91-97.
3. Derosa, L., et al., *Intestinal Akkermansia muciniphila predicts clinical response to PD-1 blockade in patients with advanced non-small-cell lung cancer*. Nat Med, 2022. **28**(2): p. 315-324.
4. Gopalakrishnan, V., et al., *Gut microbiome modulates response to anti-PD-1 immunotherapy in melanoma patients*. Science, 2018. **359**(6371): p. 97-103.
5. Matson, V., et al., *The commensal microbiome is associated with anti-PD-1 efficacy in metastatic melanoma patients*. Science, 2018. **359**(6371): p. 104-108.
6. Frankel, A.E., et al., *Metagenomic Shotgun Sequencing and Unbiased Metabolomic Profiling Identify Specific Human Gut Microbiota and Metabolites Associated with Immune Checkpoint Therapy Efficacy in Melanoma Patients*. Neoplasia, 2017. **19**(10): p. 848-855.
7. Lee, K.A., et al., *Cross-cohort gut microbiome associations with immune checkpoint inhibitor response in advanced melanoma*. Nat Med, 2022. **28**(3): p. 535-544.
8. Wang, D., et al., *Characterization of gut microbial structural variations as determinants of human bile acid metabolism*. Cell Host Microbe, 2021. **29**(12): p. 1802-1814.e5.
9. Chen, L., et al., *The long-term genetic stability and individual specificity of the human gut microbiome*. Cell, 2021. **184**(9): p. 2302-2315.e12.
10. Chen, L., et al., *Influence of the microbiome, diet and genetics on inter-individual variation in the human plasma metabolome*. Nat Med, 2022. **28**(11): p. 2333-2343.
11. Knight, R., et al., *Best practices for analysing microbiomes*. Nat Rev Microbiol, 2018. **16**(7): p. 410-422.
12. Mariathasan, S., et al., *TGF β attenuates tumour response to PD-L1 blockade by contributing to exclusion of T cells*. Nature, 2018. **554**(7693): p. 544-548.
13. Cristescu, R., et al., *Pan-tumor genomic biomarkers for PD-1 checkpoint blockade-based immunotherapy*. Science, 2018. **362**(6411).

REVIEWER COMMENTS

Reviewer #1 (Remarks to the Author):

This is a very comprehensive revision which addressed most of my comments. The paper is much improved.

1. My major comment is that if I understand it correctly, the new Pmeta are not corrected for multiple testing. While I accept the authors' justification about this being exploratory, the adjusted p-values need to be displayed so that readers can have a transparent appreciation of false discovery rate in this analysis.

2. Another significant comment relates to question 1.5. For the association between variables X and Y to be confounded by variable C variable C needs to be associated with both X AND Y. So, in the case of the association between an SV and a phenotype, we need the relative abundances to be associated BOTH with SV AND the phenotype to confound the association. Since SVs are by construction independent of relative abundances, I don't believe it matters that relative abundances are associated with the outcome. On the contrary – if the relative abundances are associated with an outcome but independent of SV, and you adjust for relative abundance when checking for associations with SV – you may be introducing an artifactual association in the back door. Bottom line, I don't think the model should control for relative abundances. I would at least confirm that the associations you found hold when you remove relative abundances from the model.

3. Question 1.1. Given the sample sizes I would have recommended defining SVs on Spencer et al. and evaluating on all the others. There are still differences between the 3 US studies (seen very well in 3c). But I think that the authors' made a reasonable analytic choice, so consider this an optional suggestion.

4. Question 1.18 – the authors mention that they remove “genetic” but this is not true. Please note the following locations:

L341 – I recommend to remove “genetic”

L343 is unclear

L344, L352,353, 354, 376, 386, etc.

5. Question 1.22 I think that sorting is not enough. Perhaps have the intensity of the color reflect effect size.

6. I'm curious as to why the number of studies changed?

7. Another optional suggestion is to draw some of the SVs as genome tracks to help with following up the results e.g. in L417+

8. Some textual stuff I picked up on while rereading the manuscript:

L64 – engaged in => collected in/from

L70 – B. caccae => Bacteroides caccae

Fig 1: “costumed designed” => custom

L181: ICI's => ICI

L787: “\$ The country that samples recruited.” => “The country in which the study was recruited”.

L262 – clarify that it's microbial genetic or use the term “SV” instead

L325-326 – Even though you analyze them together, I won't refer to them as “Combined US cohort”. These are separate studies.

Reviewer #2 (Remarks to the Author):

The authors have been highly responsive to reviewer comments and the revised manuscript is much improved. I have no further concerns.

Central South University

Response to Reviewers

Gut microbial structural variations associate with immune checkpoint inhibitor response

Nature Communications, manuscript ID: NCOMMS-23-19141A

Reviewer #1:

Question 1.0. This is a very comprehensive revision which addressed most of my comments.

The paper is much improved.

1. My major comment is that if I understand it correctly, the new Pmeta are not corrected for multiple testing. While I accept the authors' justification about this being exploratory, the adjusted p-values need to be displayed so that readers can have a transparent appreciation of false discovery rate in this analysis.

Response 1.0:

We feel moved by your effort in reviewing our manuscript so carefully and by your constructive comments on it. Thank you very much!

Thanks for your suggestion. We have added the adjusted *fd*r p-values in the supplementary Table 4.

Question 1.1. 2. Another significant comment relates to question 1.5. For the association between variables X and Y to be confounded by variable C variable C needs to be associated with both X AND Y. So, in the case of the association between an SV and a phenotype, we need the relative abundances to be associated BOTH with SV AND the phenotype to confound the association. Since SVs are by construction independent of relative abundances, I don't believe it matters that relative abundances are associated with the outcome. On the contrary – if the relative abundances are associated with an outcome but independent of SV, and you adjust for relative abundance when checking for associations with SV – you may be introducing an artifactual association in the back door. Bottom line, I don't think the model should control for relative abundances. I would at least confirm that the associations you found hold when you remove relative abundances from the model.

Response 1.1: We I thank the reviewer for raising the concern.

Central South University

We have removed the species abundance in the association analysis of the revised manuscript.

We have updated the method, the codes (https://github.com/liuronghyw/ICIs_gut_microbe_SVs), and all the results (tables S3 and S4) in this revised manuscript.

The corresponding text now reads:

Page 10-11, line 269-304:

" We examined the differences in response, PFS12, and irAEs following ICI treatment across the SV or abundance of species. The associations between OS and SV, or abundance, for each species were also investigated. The assignment values of the clinical variables were listed in Table S1C.

i) Species-level associations of the gut microbiome with clinical outcomes

The association between binary clinical outcomes and SV of each species was evaluated using PERMANOVA with 999 permutations with the following formula:

Distance matrix of SV ~ clinical outcome + Age_bins + Gender

The association between binary clinical outcomes and species relative abundance was evaluated using a logistic regression model with the following formula:

Clinical outcome ~ Species relative abundance + Age_bins + Gender

For each clinical outcomes, association analysis were performed within each cohort, and meta-analysis with a random-effect model was performed to integrate the results of different cohorts.

ii) dSV or vSV site based associations of the gut microbiome with clinical outcomes

Associations between SVs and binary clinical outcomes were assessed using logistic models with the formula:

Clinical outcome ~ SV + Age_bins + Gender

, demanding at least 10 subjects in each comparison and at least 3 responders.

Associations between SVs and OS were assessed using Cox regression models with the formula:

OS ~ SV + Age_bins + Gender

, ensuring a minimum of 20 subjects in each comparison.

In order to derive more easily interpretable HRs or ORs, quartiles (25%, 50%, and 75%) of

Central South University

the value of each vSV were computed for association analysis and modeled as continuous variables. “

Question 1.2.

Given the sample sizes I would have recommended defining SVs on Spencer et al. and evaluating on all the others. There are still differences between the 3 US studies (seen very well in 3c). But I think that the authors' made a reasonable analytic choice, so consider this an optional suggestion.

Response 1.2: Thanks for this recommendation.

As for melanoma, there are five cohorts (table 2). Given the sample sizes, Spencer et al. or Lee et al. were the candidate cohorts for defining SVs. We have tried to define SVs with Spencer et al. (n = 167) and found that SVs located in only 36 species were identified (using default parameters except --min_samp_cutoff 17). When defining SVs within a joined USA cohort (Table 1), we found that many species have SVs that can be identified with USA cohorts with a small sample size (for example, FrankelAE_2017 with 39 samples or McCullochJA_2022 with 62 samples) but can't be identified with Spencer et al. For example, SVs in *E. lenta* could be identified within 22 samples (56%) of FrankelAE_2017 and 27 samples (40%) of McCullochJA_2022, but only with 1 sample in Spencer et al. So that Spencer et al. was not suitable to be used to call SV and utilized as a reference database only. We have tried joint the cohorts from USA to call SV in the former revision.

We understand the reviewers' concern about call SVs based on "jointed USA studies". So we try to call SV with Lee et al. (n = 164), and SVs in 54 microbial species were identified. In this revision, we detected SVs in cohorts from the United Kingdom (163 samples) using SGV-Finder with default parameters (except --min_samp_cutoff 17). To analyze the replication of associations between cohorts for melanoma, we calculated for each SV region in the other four USA cohorts (FrankelAE_2017, McCullochJA_2022, PetersBA_2019, and SpencerCN_2021) its dSV or vSV in the LeeKA_2022 cohort from the UK. We run SGV-Finder with the -by-orig parameter by using the orig_dsgv.df, orig_vsgv.df, and average coverage file for each species (.df files in orig_frmaes folder) generated with the UK cohort (Table 2).

The method, results were updated in this revised manuscript. Please check it out. Thanks!

Central South University

Table 1 The number of samples with SVs identified

Short name	The number of samples with SVs identified		
	FrankelAE_2017 (Frankel et al)	McCullochJA_2022 (McCulloch et al.)	SpencerCN_2021 (Spencer et al.) [§]
E.ramulus	10	13	0
A.equolifaciens	9	12	0
C.catus	11	11	0
E.lenta	22	27	1
E.bacterium	8	13	2
B.longum	10	21	3
D.formicigenerans	19	20	4
C.bacterium	14	8	5
M.smithii	11	16	5
Collinsella sp	21	22	5
Clostridium sp	8	9	6
D.longicatena	12	13	7
B.adolescentis	9	15	7
C.clostridioforme	7	6	9
R.lactaris	10	14	9
R.gnavus	14	21	9
C.leptum	13	17	10
C.comes	20	20	10
Parabacteroides sp	9	6	11
R.torques	16	17	11
E.hallii	19	19	12
P.copri	4	5	13
Ruminococcus sp	21	33	13
B.obeum	25	31	15
B.intestinalis	9	7	16
R.hominis	13	13	16

[§] The species that with less than 17 samples of Spencer et al. were listed.

Central South University

Table 2

Study	N	Sample source	Call SV
Lee et al. Nature Medicine. 2022	164	United Kingdom	call SV with this cohort (SV_UK)
Spencer et al., Science 2021	167	University of Texas MD Anderson Cancer Center (UTMDACC) in Houston	Call SVs with SV_UK as reference
Frankel et al., Neoplasia 2017	39	University of Texas Southwestern Medical Center	Call SVs with SV_UK as reference
McCulloch et al., Nature Medicine. 2022	62	University of Pittsburgh's Hillman Cancer Center (HCC)	Call SVs with SV_UK as reference
Peters et al., Genome Medicine 2019	23	USA	Call SVs with SV_UK as reference

We have updated the whole manuscript with new analysis results, as mentioned above.

Please check it out.

Question 1.3.

Question 1.18 – the authors mention that they remove “genetic” but this is not true. Please note the following locations:

L341 – I recommend to remove “genetic”

L343 is unclear

L344, L352, 353, 354, 376, 386, etc.

Response 1.3: Thanks for your nice comments. I'm sorry for this. We have corrected the above sentence in the revised manuscript.

The corresponding text now reads:

Page 12-13, line 336-353:

“ We calculated the distance of bacterial SV profiles as described in the method between all samples within the USA or UK cohorts for melanoma (Fig. 3c). Microbial abundance (the top five PCs) can explain about 7.55% of the variance in the metagenome-wide SV profile (PPERMANOVA=0.001; Fig. S2c). After correcting for microbial abundance, the cohort

Central South University

contributed to SV differences (explaining 5.17% of the SV variance, PPERMANOVA = 0.001; Fig. S2c). Further, after correcting for microbial abundance, the SV principal coordinates (PCo) 1 and PCo2 demonstrated differences between these five cohorts (ANOVA test, $P = 1.35 \times 10^{-9}$ for PCo1 and $P = 1.34 \times 10^{-7}$ for PCo2), demonstrating a divergence of microbial SVs between these five cohorts that was independent of differences in their microbial abundances. It's interesting to note that age bins and sex, combined could only account for 0.67% of the SV profile variance in the USA and UK cohorts (Fig. S1c).

As for France cohorts for NSCLC or RCC, microbial abundance (the top five PCs) can explain about 6.11% of the variance in the metagenome-wide SV profile (PPERMANOVA = 0.001; Fig. S3C). The cohort explains 0.61% of the SV profile differences. After correcting for species abundance, there are differences between two cohorts from France of the PCo1 (ANOVA test, $P = 0.03$), and no differences was found of the PCo2 (ANOVA test, $P = 0.42$)."

Question 1.4

Question 1.22 I think that sorting is not enough. Perhaps have the intensity of the color reflect effect size.

Response 1.5: Thanks for your suggestions. I guess the following type of picture (Figure 1) is what you mean to draw.

Figure 1

However, in our heatmap, the color means the type of association, just as the figure 3A of Wang et al. did in their publications (Wang D, et al. Cell Host Microbe. 2021, Figure 2)[1].

Central South University

Figure 2

In our picture (Figure 3), purple suggests both the SV and abundance were associated with the clinical outcome. If the intensity of the color should be reflect effect size, there should be two effect size values for SV (such as the meta p value) and abundance (e.g. the meta beta value) respectively in one box of the heatmap.

Figure 3

Central South University

We conduct PERMANOVA analysis for the SV species-level associations of response, PFS \geq 12 months, and irAEs with microbial SV in each study, then combine the results with meta-analysis with the p-values and sample sizes and get the combined p values. We do not get meta effect size values (for example R^2) for SV. For abundance, we conduct meta analysis with beta values and samples sizes from each study, and get meta beta values (effect sizes) and p values for species abundance (Table S3).

Firstly, the intensity of the color can't reflect effect sizes for both SV and abundance. Secondly, we don't have meta effect size values for SV. So sorry for we can't draw picture with the intensity of the color reflect effect size. However, the meta p values for SV association, meta beta values and p values for species abundances were provided in Table S3. Thanks!

Question 1.6

I'm curious as to why the number of studies changed?

Response 1.6: Thanks for this question.

As we stated in Table S1 and Figure S1 (Figure 4, the purple box) of the former version of the manuscript, the sample size of baseline microbiome data in study PetersBA_2019 is less than 30 ($n = 23$), and the response information is lacking. PFS and OS information were available in this cohort. Since we take association analysis within each cohort, the sample size is small and lacks statistical power, so we deleted this cohort from the former manuscript.

We have added cohort PetersBA_2019 in this revised manuscript. We didn't take cox-regression analysis for OS in this cohort since we didn't take survival analysis with cohort less than 20 samples available (the sample size with SV information available is commonly less than the original sample size). We updated all the figures and tables in the revised manuscript.

Central South University

Figure 4. A flowchart of the study that details the samples utilized at each stage of statistical analysis. OS, Overall survival. OS: overall survival; PFS: progression-free survival; irAEs: immune-related adverse events. NSCLC: Non-small-cell lung cancer; RCC: Renal cell carcinoma.

Question 1.7. Another optional suggestion is to draw some of the SVs as genome tracks to help with following up the results e.g. in L417+

Response 1.7: It is a great suggestion. We have drawn some of the SVs as genome tracks in the revised manuscript figure 5 and figure 6 just as follows (generate with SGV-Finder with -- browser_path parameter). Thanks!

Central South University

Question 1.8. Some textual stuff I picked up on while rereading the manuscript:

L64 – engaged in => collected in/from

L70 – B. caccae => Bacteroides caccae

Fig 1: “costumed designed” => custom

L181: ICI’s => ICI

L787: “\$ The country that samples recruited.” => “The country in which the study was recruited”.

L262 – clarify that it’s microbial genetic or use the term “SV” instead

L325-326 – Even though you analyze them together, I won’t refer to them as “Combined US cohort”. These are separate studies.

Response 1.8: Thanks for pointing this out. We have corrected the corresponding places, as you mentioned. We use “SV” rather than “microbial genetics” throughout this manuscript. We avoid using the expression “Combined US cohort” in the revised manuscript.

1. Wang, D., et al., *Characterization of gut microbial structural variations as determinants of human bile acid metabolism*. Cell Host Microbe, 2021. **29**(12): p. 1802-1814. e5.

REVIEWERS' COMMENTS

Reviewer #1 (Remarks to the Author):

I have no additional comments.

Central South University

Response to Reviewers

Gut microbial structural variations associate with immune checkpoint inhibitor response

Nature Communications, manuscript ID: NCOMMS-23-19141B

Reviewer #1:

I have no additional comments.

Response 1.0:

We feel moved by your effort in reviewing our manuscript so carefully and by your constructive comments on it. We are so happy to know that you are willing to accept our manuscript.

Thank you very much!